# Andean drought and glacial retreat tied to Greenland warming during the last glacial period

Arielle Woods [1], Donald T. Rodbell[2], Mark B. Abbott [1✉], Robert G. Hatfield [3,4], Christine Y. Chen[5,6], Sophie B. Lehmann [1], David McGee [5], Nicholas C. Weidhaas[1], Pedro M. Tapia [7], Blas L. Valero-Garcés [8], Mark B. Bush [9] & Joseph S. Stoner[3]

Abrupt warming events recorded in Greenland ice cores known as Dansgaard-Oeschger (DO) interstadials are linked to changes in tropical circulation during the last glacial cycle. Corresponding variations in South American summer monsoon (SASM) strength are documented, most commonly, in isotopic records from speleothems, but less is known about how these changes affected precipitation and Andean glacier mass balance. Here we present a sediment record spanning the last ~50 ka from Lake Junín (Peru) in the tropical Andes that has sufficient chronologic precision to document abrupt climatic events on a centennial-millennial time scale. DO events involved the near-complete disappearance of glaciers below 4700 masl in the eastern Andean cordillera and major reductions in the level of Peru's second largest lake. Our results reveal the magnitude of the hydroclimatic disruptions in the highest reaches of the Amazon Basin that were caused by a weakening of the SASM during abrupt arctic warming. Accentuated warming in the Arctic could lead to significant reductions in the precipitation-evaporation balance of the southern tropical Andes with deleterious effects on this densely populated region of South America.

[1] Department of Geology and Environmental Science, University of Pittsburgh, Pittsburgh, PA, USA. [2] Geology Department, Union College, Schenectady, NY, USA. [3] College of Earth, Ocean, and Atmospheric Science, Oregon State University, Corvallis, OR, USA. [4] Department of Geological Sciences, University of Florida, Gainesville, FL, USA. [5] Department of Earth, Atmospheric, and Planetary Sciences, Massachusetts Institute of Technology, Cambridge, MA, USA. [6] Division of Geological and Planetary Sciences, California Institute of Technology, Pasadena, CA, USA. [7] Instituto Nacional de Investigación en Glaciares y Ecosistemas de Montaña, Ancash, Peru. [8] Pyrenean Institute of Ecology, Spanish National Research Council, Zaragoza, Spain. [9] Florida Institute of Technology, Melbourne, FL, USA. ✉email: mabbott1@pitt.edu

Variations in Atlantic Meridional Overturning Circulation (AMOC) during the last glacial cycle drove abrupt changes in the thermal gradient of the North Atlantic sector, altering the interhemispheric distribution of tropical heat, the mean position of the intertropical convergence zone (ITCZ), and trade wind strength[1–3]. South American low-latitude paleoclimate proxy records are sensitive to high-latitude forcing via the strength of the South American summer monsoon (SASM), which increased during cold stadial periods such as Heinrich events[4–7], and weakened during the abrupt warmings recorded in Greenland ice cores associated with Dansgaard–Oeschger (DO) interstadials[5,8,9].

Large fluctuations in Andean paleolake levels have been documented on the Bolivian Altiplano in association with the Younger Dryas and Heinrich events 1 and 2[7,10,11], and Andean glacier advances or stillstands have been linked to these wet events in some cases[12]. Less is known about the effects of the shorter duration DO cycles on precipitation anomalies and on the mass balance of tropical Andean glaciers. Although some studies proposed a causal link between local glacier fluctuations and the sedimentary record of Lake Titicaca[13], well-dated continuous records are necessary to test this hypothesis. Much of the paleoclimatic evidence documenting these rapid changes in tropical South American hydroclimate relies on the interpretation of $\delta^{18}O$ variations in speleothems from the Amazon Basin and surrounding regions[5,6,8]. Similarities among speleothem $\delta^{18}O$ records reflect the regional impact of variations in convective activity and upstream rainout in the core monsoon region of Amazonia[14]. However, speleothem records from several localities do not reveal a tight coupling between independent proxies of local precipitation amount and the $\delta^{18}O$ of that precipitation ($\delta^{18}O_{precip}$)[15,16], indicating that upstream factors other than the amount effect[17] may dominate $\delta^{18}O_{precip}$ at some locations. The inability to isolate local precipitation variations from the composite $\delta^{18}O$ signal[18,19] makes it difficult to assess the specific impact of abrupt Arctic warming on water availability and glacial mass balance in the tropical Andes, and it highlights the need for $\delta^{18}O$-independent records of hydroclimate.

Here we present a sediment record from a high elevation lake in the central Peruvian Andes that records lake level fluctuations as well as changes in paleoglacier mass balance. We show that the DO interstadials between 50 and 15 ka, which are recorded isotopically both in Greenland ice[20,21] and speleothem $\delta^{18}O$ from Pacupahuain Cave in the upper Amazon Basin[5], were associated with rapid and large reductions in Andean precipitation amount recorded by multiple independent proxies in Lake Junín sediments. Many of these perturbations were sufficient to deglaciate the adjacent portion of the eastern Andean cordillera up to at least 4700 masl and profoundly shrink Lake Junín, Peru's second-largest lake located at 4100 masl and ~25 km from Pacupahuain Cave (Fig. 1). This record documents the unambiguous impact on glacier mass balance and hydroclimate of the climatic teleconnection linking the Atlantic meridional thermal gradient with the strength of the SASM over the past 50 ka.

## Results

**Lake setting and glacial connection.** Lake Junín (11°S) is a seasonally closed-basin lake located between the eastern and western cordilleras of the central Peruvian Andes (Fig. 1). With a surface area of ~280 km[2] and a seasonally variable water depth of ~8–12 m, Lake Junín is especially sensitive to changes in precipitation–evaporation balance (P–E). The watershed occupies the Puna grasslands ecoregion where groundwater-fed peatlands (*bofedales*), characterized by organic-rich sediment, occupy the shallow water lake margins. Glacial outwash fans and lateral

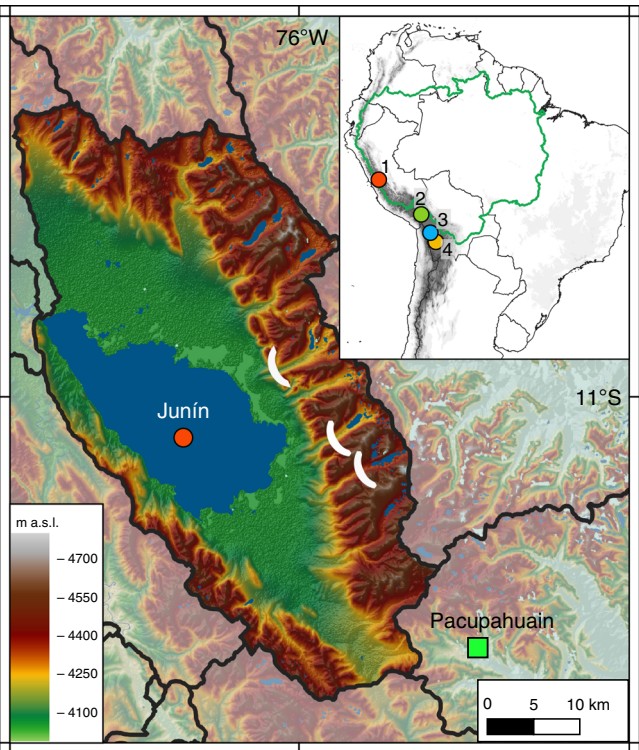

**Fig. 1 Location of the Lake Junín (4100 masl) drainage basin and Pacupahuain cave in central Peru.** White ellipses in valleys east of Lake Junín indicate the mapped down valley extent of glaciers during the local Last Glacial Maximum[24]. Inset map: green line indicates the Amazon drainage basin, and circles indicate Andean records discussed in the text: 1. Junín, 2. Lake Titicaca[13], 3. Paleolake Tauca[10,12], 4. Salar de Uyuni[7].

moraines form the basin's eastern and northern edges (Fig. 1), and [10]Be exposure ages from these moraines indicate they span multiple glacial cycles[22,23], but at no time during at least the last 50 ka has the lake been overridden by glacial ice[24]. Thus, Lake Junín is ideally situated to record the last glacial cycle in the adjacent eastern cordillera. During the local last glacial maximum (LLGM; ~28.5–22.5 ka) alpine glaciers descended from headwall elevations as high as ~4700 masl to end moraine positions ~4160 masl, within several km of the modern shoreline[24]. Whereas glaciers in the inner tropics of the Andes are especially temperature-sensitive because of sustained precipitation year-round, glaciers in the outer tropics, such as those at the latitude of the Junín basin, experience greater seasonality of precipitation and are twice as sensitive to changes in precipitation as those in the inner tropics[25,26]. The Junín region receives most of its moisture through the SASM during the austral summer (DJF) with <7% falling during the winter (JJA), making variations in the SASM a principal driver of changes in paleoglacier mass balance.

Most records of glaciation in the tropical Andes rely on moraine exposure ages to infer the timing and extent of advances[12,23,27]. However, such records are limited by age uncertainties of ~±5%, an unknown temporal relationship between the timing of moraine stabilization and ice advance, and the tendency for larger advances to erase evidence of prior glacial cycles. Continuous proxy records from well-dated glacier-fed lakes such as Junín can compensate for such limitations, with clastic sediment flux and high-resolution XRF scans being well-established proxies for glacial erosion of bedrock that, in turn, reflect relative changes in paleoglacier activity and mass balance[28,29]. Accordingly, final deglaciation of the Junín

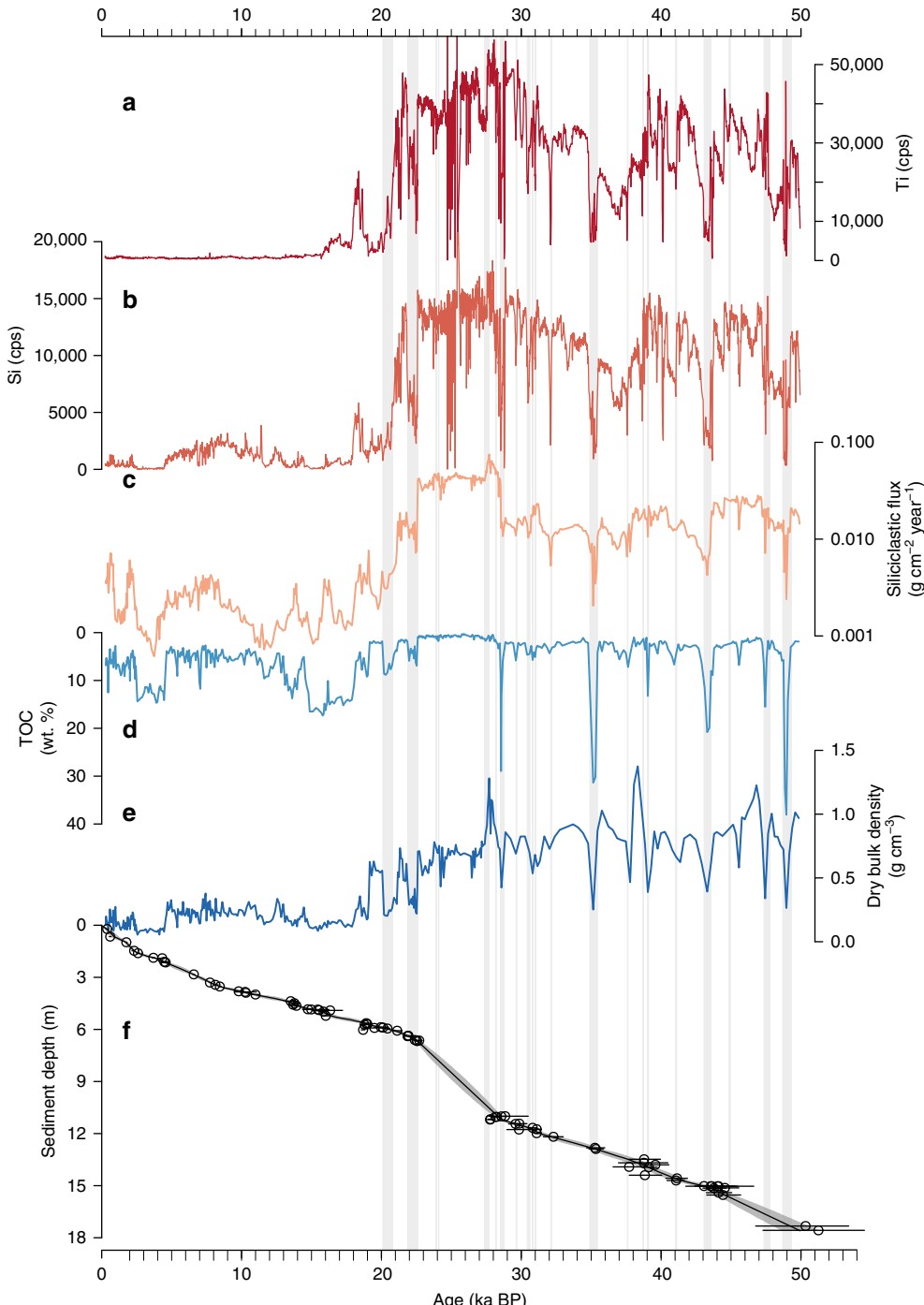

**Fig. 2 Physical and geochemical sediment properties from the Junín drill core.** The similar XRF profiles of **a** Ti and **b** Si indicate both elements primarily represent clastic inputs, with slight differences attributable to different bedrock mineralogy and grain size. **c** Siliciclastic sediment flux (log scale). **d** Total organic carbon (TOC). **e** Dry bulk density. **f** Bacon age-depth model of 79 AMS radiocarbon ages on terrestrial macrofossils; error bars denote 95% probability range for calibrated ages. Ages excluded from the age model are listed in the supplemental. Gray vertical bars show the distribution of peat layers.

watershed was nearly complete by 18 ka and was marked by a near-total cessation of clastic sediment input to the lake[29,30].

**Sedimentary context**. The Junín sediment cores were obtained from the lake depocenter (Fig. 1) in 8.2 m of water. The age model for the last 50 ka (cal yr BP, 1950 CE) is based on 79 radiocarbon measurements from terrestrial macrofossils and charcoal (Fig. 2f and Supplementary Table 1). Sediment deposited from 50–22.5 ka is dominated by fine-grained glacial flour

characterized by high Ti and Si counts per second (cps), high density, and low total organic carbon (TOC) (Fig. 2a–e). Glacigenic sediment input to Lake Junín was especially high and devoid of plant macrofossils from 28.5–22.5 ka (Fig. 2a–c), which corresponds to the age of moraines deposited during the maximum extent of ice in the last 50 ka in the adjacent eastern cordillera[23], according to [10]Be ages recalculated based on updated production rates and scaling factors[31]. The scaling procedure applied here[31] yields [10]Be ages that are compatible within ±2%

with those that would be obtained using local tropical Andes [10]Be production rates[32,33]. Glacigenic sediment deposition was punctuated by a series of distinct 1–20-cm-thick peat layers (Fig. 2) containing 5–35% TOC (Fig. 2d) with abundant macrofossils that are similar to the sediment accumulating today in the fringing peatlands. These peat layers span intervals from ~25–500 years based on mean sedimentation rates and are interpreted to reflect lake low stands when surrounding peatlands encroached toward the center of the lake, forming a record of repeated water level fluctuations of up to ±8 m. There is no sedimentary or radiocarbon evidence (Fig. 2f) for unconformities within age model precision, which indicates that while these peat layers represent considerably lower water level, the drill site remained submerged, at least seasonally, for the duration of our record. Today Lake Junín overflows during the summer wet season and there is no evidence of shoreline features above the modern lake level. Sediment deposited after ~20 ka reveals a rapid decline in clastic input, evidenced by the low values of Ti, Si, bulk density, and siliciclastic flux. The near-complete loss of the clastic signal does not imply reduced precipitation after 20 ka, but rather reflects a different sedimentary environment wherein glaciers had retreated behind up valley moraine-dammed lakes that served as local sediment traps. During the late glacial, the sedimentary regime shifted to a lake increasingly dominated by authigenic $CaCO_3$ with frequent laminations (Fig. 2c–e and Supplementary Fig. 1). The occasional fine-grained organic-rich intervals <20 ka contain few plant macrofossils and record autochthonous (algal) productivity; these intervals do not resemble the dark, crumbly, and macrofossil-rich peats that punctuate the 20–50 ka interval of the core.

**Synchronous changes in lake level and paleoglacier extent**. The Junín record exhibits a reduced input of glacigenic sediment during DO interstadials 4–13 (Fig. 3a), with all but two of these intervals marked by enhanced peat accumulation, associated higher TOC, and lower density (Fig. 2d, e). Similar but less pronounced changes appear to be associated with DO interstadials 2 and 3. However, the timing of these latter events coincides with the period of rapid clastic sedimentation during the LLGM, an interval that is devoid of datable macrofossils and therefore has less robust age control (Fig. 2f). Declines in siliciclastic sediment flux (Fig. 2c) indicate that simple dilution effects were not responsible for the reductions in glacigenic sediment concentration. The timing of DO interstadials was thus marked by widespread glacial retreat and lake level lowering up to ~8 m, within the chronologic uncertainty of our age model (Supplementary Fig. 2). The absence of evidence for lowered lake level during complete and final deglaciation of the Junín catchment (~20–15 ka), when regional warming drove snowlines to rise 300–600 m[24,30], indicates that the level of Lake Junín is especially sensitive to precipitation amount rather than variations in temperature. The close association between low lake stands and reduced glacial sediment flux during DO events suggests that these reductions in paleoglacier mass balance were primarily driven by decreases in precipitation. The declines in lake level associated with the DO events noted here corroborates the evidence of water level reductions associated with DO interstadials 11, 10, and 8 at 1360 masl in southern Peru (14°S)[34], and evidence for millennial-scale fluctuations in nearshore terrigenous inputs to Lake Titicaca that may be linked to DO events[13]. The documented changes in hydroclimate in the Junín region may thus have affected a large region of the westernmost Amazon Basin, which is consistent with the Fe/Ca record of Amazon River discharge[9] (Fig. 3e). Relative to Altiplano lake records[7,10], Junín most strongly records dry (DO) events, whereas the Altiplano records are more sensitive to wet (Heinrich) events. Thus, the records appear to be biased toward recording hydroclimatic perturbations that contrast the most with their respective background states; namely, more arid on the Altiplano and more humid in the Junín region.

**Discussion**
On millennial timescales, multiple independent proxies measured on Junín sediments bear a strong correspondence to the precisely dated speleothem $\delta^{18}O$ records from both the nearby Pacupahuain Cave[5] (Fig. 3c) and from El Condor Cave (Fig. 3d), a lower elevation site (800 masl) in the western Amazon Basin of northern Peru[8]. This concurrence indicates that regional monsoon strength was a first-order control on all records. While downcore changes in clastic sediment inputs at Junín are indicative of relative, rather than absolute, changes in precipitation, they record an entirely local signal. The observed similarity with nearby speleothems suggests that $\delta^{18}O_{precip}$, which has been interpreted to reflect upstream convection and rainout[5,14,35], also reflects some degree of variable local precipitation amount in the tropical Andes. However, the magnitude and duration of Junín's response to individual DO events is often not to scale with that of Pacupahuain, only 25 km away. For example, DO interstadials 11 and 13 register as profoundly dry intervals at Junín but only minimally so in Pacupahuain, contrary to the signal that would be predicted by a simple amount effect[17]. A similar mismatch occurs during DO interstadial 8, which is a relatively weak dry period at Junín with moderate reductions in Ti and Si and only a multi-decadal interval of peat accumulation, yet DO 8 in the Pacupahuain record is marked by the most positive $\delta^{18}O$ excursion in the entire speleothem sequence, lasting nearly a millennium. These observations indicate that the local moisture response at Junín can be disproportional to, and possibly even decoupled from, the $\delta^{18}O$ signal that is thought to be recording millennial-scale SASM intensity. This confirms earlier work showing that atmospheric transport of water vapor from the tropical Atlantic across the Amazon lowlands involves numerous isotopic controls, in addition to precipitation amount, which influences the $\delta^{18}O_{precip}$ signal of geologic archives[18,19,35].

The early onset of deglaciation in the central Peruvian Andes, ~22.5 ka based on lake sediment records[36] (Fig. 3a), is consistent with moraine ages that reflect retreating ice margins at this time[23,27]. This onset was several millennia prior to the onset of global deglaciation as recorded by sea-level rise[37] (Fig. 3g), and was initially interpreted as evidence for early tropical warming because of the lack of evidence for drying at this time[36]. The Junín peat record, however, reveals that two prolonged droughts, lasting a total of ~1300 yr, occurred in quick succession (22.5–21.9 ka and 20.8–20.1 ka), just prior to the onset of warming ~20 ka in the high latitudes of the Southern Hemisphere[38] (Fig. 3h). We suggest that these prolonged dry intervals, which resemble Junín's response to DO events, were responsible for the early onset of glacial retreat in this region of the tropical Andes. These abrupt reductions in precipitation at Junín are evident, though subtle, in the Pacupahuain record, yet they do not appear as pronounced individual excursions in AMOC[1] or Amazon discharge[9] (Fig. 3e, f). It is notable, however, that the latter two records indicate that the period from ~24 to 19 ka was characterized by a relatively strong AMOC and overall drier conditions in the Amazon Basin, respectively. These observations, along with records of tropical Atlantic mixed layer depth[3], indicate that the 24–19 ka interval was not marked by the large southward ITCZ displacements that characterized Heinrich events 2 and 1, and this may explain why Junín experienced extended droughts and early deglaciation during this interval.

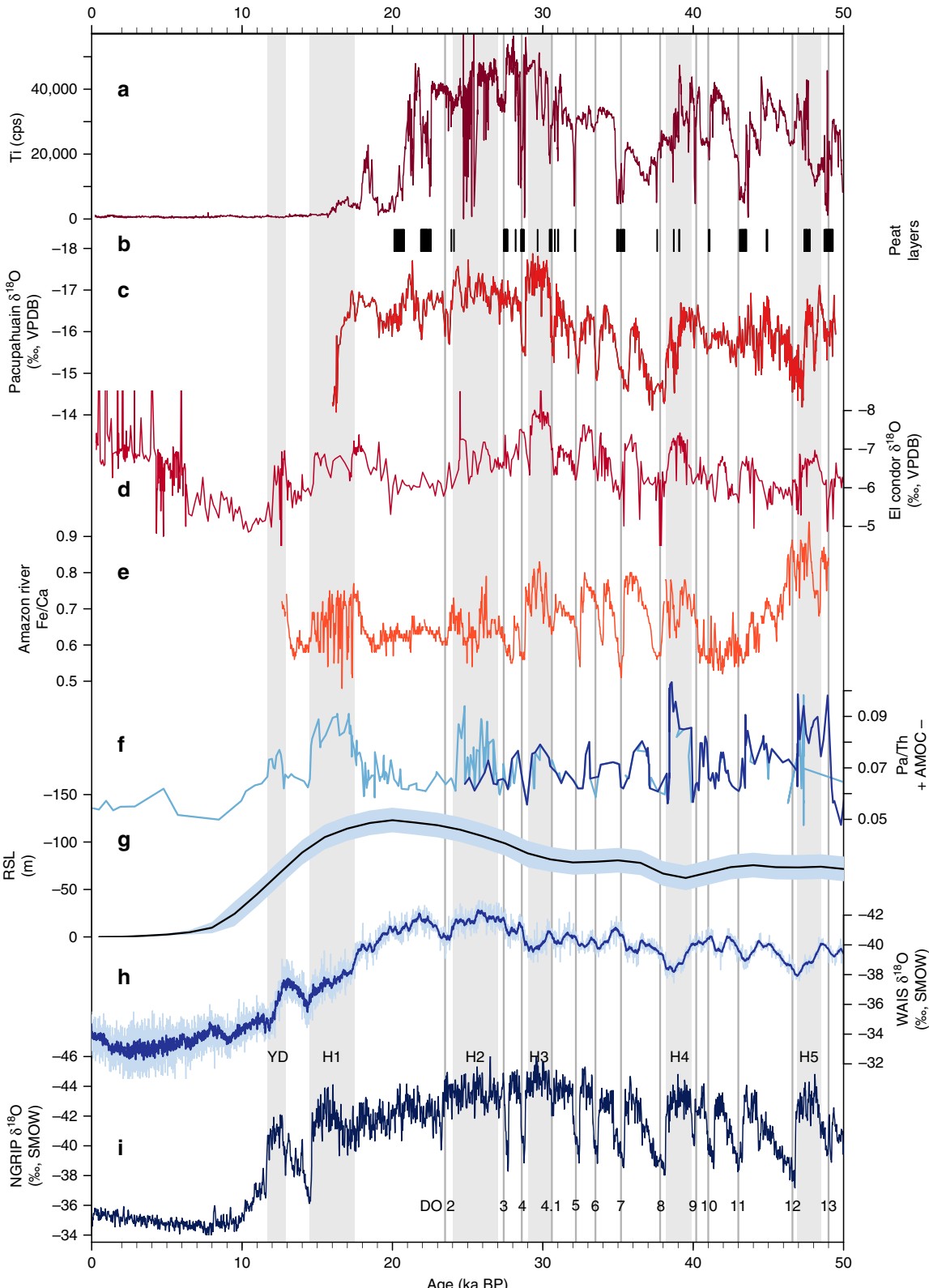

**Fig. 3 Comparison of regional and global proxy paleoclimatic records. a** Junín glaciation (Ti from Fig. 2a). **b** Junín low stands (peat layers). **c** Pacupahuain speleothem δ18O[5]. **d** El Condor speleothem δ18O[8]. **e** Amazon Discharge[9]. **f** Atlantic Meridional Overturning Circulation (AMOC) strength (dark blue curve is Pa/Th data reported in (1)), and the light blue curve is a compilation of previously reported Pa/Th records as presented in (1). **g** Relative sea level[37]. **h** WAIS Divide ice core δ18O[38]. **i** NGRIP ice core δ18O[20,21]. Vertical gray boxes denote the Younger Dryas and Heinrich stadials H1–H5, and numbered vertical lines are Dansgaard–Oeschger (DO) warming events 2–13.

Alternately, modeling studies have pointed to a thermo-dynamically driven contraction of the tropical rain belt associated with global cooling during the global LGM[39], which may have contributed to reductions in SASM rainfall and early deglaciation in the tropical Andes.

The significant disruption to glaciers and hydroclimate in the tropical Andes in response to perturbations in the meridional temperature gradient of the North Atlantic documented here demonstrates the sensitivity of tropical P–E balance to the Northern Hemisphere climatic shifts. There are multiple possible scenarios for regional hydroclimatic change in the Amazon Basin in response to twenty-first-century warming. One scenario posits that accentuated warming in the Arctic will result in a northward shift in the mean position of the ITCZ[40,41], while another projects a stable mean position of the ITCZ, but reductions in both width and strength[42]. Past changes in the hemispheric thermal gradient are an imperfect analog for future change, however. For example, the attendant increase in AMOC strength documented during warm interstadials is not expected to accompany future $CO_2$-induced warming[43], and changes in ITCZ position could be superimposed[41] on enhanced tropical precipitation that is pro-jected by some models[44]. Nevertheless, a northward shift in the thermal equator remains likely given faster heating of the Northern Hemisphere, and this scenario is consistent with a multimodel ensemble of simulations which projects regional drying of Amazonia and much of the tropics[45]. This would lead to significant reductions in P–E in the tropical Andes with impacts on glaciers, water supplies, hydropower, and agriculture in a region inhabited by tens of millions of people.

## Methods

**Chronology**. Samples and standards for radiocarbon age determination were chemically pretreated, vacuum sealed, and combusted at the University of Pittsburgh according to standard protocols (https://sites.uci.edu/keckams/files/2016/12/aba_protocol.pdf), and graphitized and dated by accelerator mass spectrometry at the W.M. Keck Carbon Cycle AMS facility at the University of California, Irvine. Radiocarbon samples obtained from off-splice core sections were assigned stratigraphically equivalent splice depths; the development of the splice is detailed elsewhere[46]. Radiocarbon age measurements with errors and median calibrated age estimates are reported in Table S1. An age model was constructed based on 79 radiocarbon ages spanning the upper 17.6 m of sediment. The IntCal 13 calibration curve[47] was used to calibrate all dates <50,000 cal yr BP, and the CalPal2007_Hulu calibration curve[48] was used for those >50,000 cal yr BP. Bayesian age-depth modeling was performed using the R software package Bacon v. 2.3[49] with the following settings: acc.mean = 25 yr cm$^{-1}$, acc.shape = 1.5, mem.strength = 4, mem.mean = 0.7, and thick = 5 cm.

Initial age model test runs indicated that in some instances the mean age estimates were biased toward outliers at the expense of clusters of more consistent dates. Therefore, dates that fell outside of the model's 95% confidence interval were excluded prior to final age model construction if they also met other criteria, such as small sample mass (<1 mg) or low $CO_2$ gas yield (<10 Torr) when a larger sample was available from the same interval, or if a reversal was evident based on multiple surrounding dates. In addition, the two oldest measured dates that are within the IntCal 13 calibration range (UCIAMS# 193160, 193169) would imply that sedimentation rates from ~45–50 ka were higher than any other portion of the record, including the LGM. We suggest these two ages are erroneously young and warrant exclusion. The stratigraphy does not indicate a change in sedimentation regime during this interval, and there is a pair of older dates between them (calibrated using the CalPal2007_Hulu curve) that are in close stratigraphic and chronologic agreement. Finally, sample 201045 yielded the largest lab error and is closest to the limits of the radiocarbon method, and it represents an age-reversal of several thousand years in the context of three underlying samples. Dates excluded from the final age model are marked by an asterisk in Supplementary Table 1. Calibrated median ages and 95% probability ranges are rounded to the nearest 5 yr for ages <1000, the nearest 10 yr for 1000–10,000, the nearest 50 yr for 10,000–20,000, and the nearest 100 yr for >20,000.

**Physical analysis of sediment cores**. Bulk density was determined from the air-dried mass of 1 cm$^3$ sample taken every 2.5 cm above 6.665 m and every 8 cm below 6.665 m. Total organic and inorganic carbon (TOC, TIC) was measured on samples taken every 2.5 cm above 6.665 m and every 4 cm below 6.665 m. Total carbon (TC) was determined by combusting samples at 1000 °C using a UIC 5200 automated furnace, and analyzing the resultant $CO_2$ using a UIC 5014 coulometer

at Union College. Similarly, TIC was determined by acidifying samples using an Automate acidification module and measuring the resultant $CO_2$ by coulometry at Union College. We calculated the weight percentage TOC from TOC = TC − TIC. We measured the biogenic silica (bSiO$_2$) content of a total of 65 samples obtained randomly from all facies present in the sediment core. Siliciclastic flux (Flux$_{clastic}$, Eq. 1) was calculated as:

$$\text{Flux}_{clastic} = SR*(BD - ((BD*TOM) + (BD*TCC))), \quad (1)$$

where SR is bulk sedimentation rate (cm yr$^{-1}$), BD is bulk density (gm cm$^{-3}$), TOM is the weight fraction organic matter of the bulk sediment, and TCC is the weight fraction CaCO$_3$ of the bulk sediment. We calculated TOM from TOC (%)/44 to reflect the molar ratio between plant cellulose ($C_6H_{10}O_5$)$n$ and TOC (%), and we calculated TCC from TIC (%)/12 to reflect the molar ratio between TIC (%) and CaCO$_3$. Because of the presence of both authigenic and detrital CaCO$_3$ in the sediment core, the removal of all CaCO$_3$ in the estimation of Flux$_{clastic}$ results in an underestimation of the total detrital flux during intervals of high clastic input, and this, in turn, reduces the amplitude of change in clastic flux between intervals of high and low glacigenic sediment input. While we do not explicitly remove the bSiO$_2$ in the calculation of Flux$_{clastic}$, the average weight percentage bSiO$_2$ for all 65 samples facies samples is 0.92 ± 1.12% (±1σ), and thus is negligible.

XRF scanning was performed at the LacCore XRF Lab, University of Minnesota-Duluth Large Lakes Observatory using a Cox Analytical ITRAX with a Cr tube, 5 mm resolution, and 15 s dwell time.

## Data availability

All data sets generated during the current study are available on the NOAA World Data Service for the Paleoclimatology archive at https://www.ncdc.noaa.gov/paleo/study/31152.

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

## Acknowledgements

We recognize the late G.O. Seltzer for his seminal work on the Junín Plain. We are grateful to Lake Junín Drilling Project members for their contributions to fieldwork and data collection, the International Continental Drilling Program (ICDP) for financial and logistical support, and DOSECC Exploration Services and Geotec Peru for drilling expertise. We thank LacCore for access to facilities, core curation, XRF analyses, and data management. This research was supported by grant 02–2012 from the ICDP and awards EAR-1404113 (Abbott), EAR-1402076 (Rodbell), EAR-1400903 (Stoner), EAR-1404414 (McGee), and EAR-1402054 (Bush) from the U.S. National Science Foundation Paleo Perspectives on Climate Change Program.

## Author contributions

D.T.R. and M.B.A. conceived the study. A.W., D.T.R, and M.B.A. wrote the manuscript. A.W. performed radiocarbon analyses and data interpretation. A.W., D.T.R., M.B.A., R.G.H., C.Y.C., N.C.W., and P.M.T. contributed to the fieldwork campaign. A.W., D.T.R., M.B.A., R.G.H., C.Y.C., S.B.L., D.M., N.C.W., P.M.T., B.L.V.-G., M.B.B., and J.S.S. discussed the results and provided manuscript feedback.

## Competing interests

The authors declare no competing interests.
