## [Peer Review File · Nature Communications]

Reviewers' Comments:

Reviewer #1:

Remarks to the Author:

The manuscript by Woods et al. describes a high-resolution record of landscape history from Lago Junin in the tropical Andes over the last 50k years and its relationship to Dansgaard Oeschger variability, first identified in the Greenland ice core data. The data are of very high quality, and the major results confirm prior studies on millennial variability in tropical South America in documenting the waxing and waning of Andean glaciers over this period in response to variation in precipitation driven by the South American Summer Monsoon (SASM). The results are similar to those of the Lake Titicaca record (Fritz et al. 2010), which is not as well dated as the Lago Junin record presented in this paper and does not have the high temporal resolution provided by scanning XRF. Yet, it clearly shows millennial (D-O) variation in the period between 60 and 20 ka, and this variability was attributed to differential input of clastic sediments, because of variation in precipitation (SASM) and variation in clastic material from precipitation sensitive Andean glacier advance and retreat. In addition, several speleothem records from the eastern slope of the Andes document SASM variation associated with D-O variability with high temporal precision, as acknowledged in this manuscript. The submitted manuscript in its current form overstates the novelty of the results and has some limitations in scope and interpretation that I summarize below.

Woods and colleagues suggest that there are ambiguities in speleothem records of hydroclimate in terms of the relationship between local precipitation amount and $\delta^{18}O$, necessitating additional proxies, such as lake records (line 51-57), to validate interpretations. It is true that $\delta^{18}O$ records are influenced by multiple variables and hence are not always unambiguous recorders of precipitation amount. Yet the two studies that they cite from South America (Wortham, Ward, references #11,12) show coherent signals in $\delta^{18}O$ records from sites within the heart of the SASM, such as the Andean highlands and slopes, and do not suggest that alternative interpretations of the original publications are warranted for speleothem $\delta^{18}O$ results from this region of tropical South America. Thus, both cited studies interpret the speleothem records from the Andes and western Amazon as effective recorders of changes in SASM intensity and local precipitation patterns. The sites where $\delta^{18}O$ and Sr/Ca reconstructions show some divergence, and where the Wortham and Ward papers postulate that moisture sources in addition to monsoon-sourced precipitation might be important, are all in the eastern Amazon and southeastern Brazil, not the Andes. These regions have somewhat more complex precipitation influences (e.g. from the SACZ). It is also noteworthy that these studies are Holocene in age, a period when the SASM monsoon was likely not as strong as during the cold stadials of MIS2 and MIS3.

It is also true that clastic flux and the associated geochemical signals in a sediment core are influenced by multiple variables, and thus these are not unambiguous recorders of precipitation or of glacial mass balance as suggested (line 66). The authors treat the Si and Ti signals as semi-quantitative records of precipitation (lines 150-159), when they are not; they are qualitative signals of wet versus dry. Thus, there is no reason that the magnitude of $\delta^{18}O$ variation in the nearby Pacupahuain speleothem record and the clastic signal in Lake Junin should scale equally in response to the same precipitation forcing. The assumption that the Lago Junin record is the real measure of local precipitation (lines 157-162) is not well founded, and the associated inference that the speleothem records are more ambiguous recorders of local precipitation amount than the record from Lago Junin is not supported.

The paper makes no comparison of the Lago Junin record to the nearby Lake Titicaca record (Fritz et al. 2010), despite the many similarities between the two records and the similar focus of the earlier

publication on Lake Titicaca. There are clear advantages to the Lago Junin record, such as the greatly improved age control throughout and the highly resolved record. Yet the paper does not take the opportunity to build on prior foundational studies or to acknowledge the nearby Lake Titicaca record of water level and climate change associated with DO events (e.g. lines 136-140).

It is noteworthy that the paper says nothing about Heinrich events, despite considerable evidence in tropical South America from speleothems, lake cores, and marine cores that these were among the largest excursions in the hydroclimate of tropical South America during MIS3-2. The absence of a clear signal of the Heinrich events in Lago Junin warrants discussion given the temporal focus of this paper.

A last paragraph that emphasizes the new insights from the Lago Junin record itself would improve the manuscript. Multiple prior records and publications have documented the sensitivity of tropical P-E balance to Northern Hemisphere climate perturbations. Similarly, there are a number of detailed modeling studies about the contemporary mass balance of glaciers in the tropical Andes and their potential response to 21st century warming that provide more insight about the potential future fate of regional glaciers and water supply.

In summary, I think a reframing of various aspects of the manuscript, as suggested above, would help to showcase the high-quality results and generate a more effective addition to the literature on Andean paleoclimate variability.

Sheri Fritz, University of Nebraska - Lincoln

Reviewer #2:

Remarks to the Author:

This manuscript by Woods et al. presents a truly fascinating and unique study on glacial and deglacial climate in the tropical Andes, using continuous and high resolution sediments from Lake Junín drill cores. The paper argues for a connection between warm interstadials in Greenland and moisture balance, as well as glacier mass balance, in the tropical Andes. The mechanism is that DO warm events in Greenland correspond to higher TOC and the formation of peat at/near the core location, along with a decrease in siliciclastic flux, all of which indicate dry intervals in the Junín watershed. Due to the sensitivity of glaciers in this part of the Andean tropics to precipitation (rather than just temperature), the authors further argue that these dry periods also correspond to decreases in local glacier mass balance. These results yields two high impact implications: First, that the amount of precipitation in the Junín catchment is indeed tied to South American Monsoon variability, just as studies based on the $\delta^{18}O$ of precipitation (ice cores, speleothems) have found, but only generally speaking; the relationship is not 1:1 because of local factors that may enhance or decrease precipitation in the absence of corresponding variations in the SASM. So, variations in $\delta^{18}O$ should not necessarily be interpreted as a % increase/decrease in local precipitation amount although the overall relationship holds. Second, this paper presents a new mechanism for the "early deglaciation" previously observed in the tropical Andes: That precipitation-sensitive glaciers were first diminished by drought, and then were further diminished by deglacial warming.

The paper is novel and exciting, and presents some really special and new food for thought for the paleoclimate community. It is certainly worthy of publication in Nature Communications. However, there are some important issues that need to be resolved before the manuscript can merit publication. The major issues are outlined below, followed by minor issues.

Major comments:

1) Age model. The number of radiocarbon dates for this record is truly impressive so I hate to provide negative feedback here, however there is a large gap in ¹⁴C dates submitted between 22,600 kyr BP and 28,900 kyr BP. This is problematic because it questions the evidence for the following claims that the authors make:

- "Glaciogenic sediment input to Lake Junín was especially high from 28.5-22.5 ka, which corresponds to the age of moraines deposited during the maximum extent of ice in the last 50 ka in the adjacent eastern cordillera" (lines 106-109). The timing of the high glaciogenic input is questionable here. All that is clear is that it was high at some point before 22.5 ka and after 28.5 ka. But it is quite possible that sedimentation rates proceeded at their pre-28.5 ka values until 23 ka or 24 ka or any other time within that interval. Assuming that the sed rate changed essentially right after the last ¹⁴C date prior to the interval seems unfair.

- "There is no evidence... for unconformities" (lines 115-116). In fact there is either a very large sedimentation rate change that just so happens to correspond to 28.5 ka (as suggested in the lines referenced above) or there are hiatuses, or both. It seems extremely unlikely that an 8m deep lake would come close enough to drying out that it leaves behind peat deposits, but manages somehow to not dry out completely.

- "The Junín record exhibits a reduced input of glaciogenic sediment during DO interstadials 3-13" (line 124). DO interstadial 3 is suspect because of the lack of age control in the Junín record at this time.

It seems that the gap in age control points is between cores D15-D9 and 1B-4H-1. If there is some reason for this (is this interval devoid of macrofossils?) then the authors need to be very clear about that and need to explain how the lack of age control in this important interval might complicate their interpretations of climate during the LLGM.

2) Overall relationships between glaciogenic influx and lake level. Clearly the glacial period and the Holocene are two very distinct regimes in this lake, with the glacial period being characterized by high inputs of Ti, Si, and siliciclastic flux but a very different regime once the catchment was fully deglaciated. The reason that lake levels were high during the glacial period is not the same reason they are high today (which is actually not clear—why are they so high today anyway??) If a non-paleolimnologist were to read this paper there would be confusion about why you cannot look at high Ti during the glacial period and infer high lake levels, but then look at low Ti during the Holocene and infer low lake levels (since the lake level today is quite similar to the highest highstands during the glacial period). To this end I suggest:

- Devote some text in the introduction to a clear description of the physical processes by which higher ice volume leads to higher glaciogenic sediment/siliciclastic flux. In addition to the confusion mentioned above it is a little counter-intuitive that one would not associate high glaciogenic flux with warm intervals, for example (and subsequent melting of glaciers in the catchment leading to higher runoff to the lake). I can assume some of the physical reasoning behind this but it needs to be explicitly stated in the manuscript. If there are substantial lead or lag times between glacier advance and glaciogenic input this needs to be stated and discussed too.

- Consider adding some notation to Figure 2 that shades or otherwise denotes the deglaciated portion of the record.

- Consider adding TIC data to Figure 2 since the text says that carbonate increases substantially after deglaciation and, presumably, one would interpret variations in TIC vs. TOC vs. siliciclastic flux quite

differently.

3) Figure clarity: The timing of events in the records are rather hard to read because of the tall orientation of Figures 2 and 3 and the small tickmarks on the X-axis. Two suggestions:

- Repeat the X axis on top of the figure so that it is easier for readers to figure out the timing of events in the different timeseries
- Move figure 2f to a separate panel within the figure, giving Figure 2a-e its own set of X axes (top and bottom). Then, add triangles to the top X axis denoting where age control points occur (as is common with speleothem papers)
- In Figure 3, shade or otherwise denote the time periods toward the end of the glacial period where the peat deposits occur (roughly 22.5-20.1 ka) and where the "dry intervals cause early onset of glacial retreat" argument is rooted (lines 170-172)

Minor comments:

- Line 39: Change this to "South American low-latitude paleoclimate proxy records" since this sentence talks about the SASM and not low latitude records in general
- Lines 75-78: How do we know that "at no time during at least the last 50 ka has the lake been overridden by glacial ice" ? Is there no evidence for moraines along the southwestern shore in Fig 1?
- Lines 85-87: Provide a citation for the "less than 7% falling during the winter (JJA)" number
- Throughout manuscript: Shouldn't the term be spelled glaciogenic, not glaciagenic?
- Lines 130-133: This statement is not very clear. What time period exactly are they referring to as the "late glacial-to-Holocene transition" ? It is confusing because TOC does increase during the deglaciation, unlike what they are saying here, but perhaps they are referring to some specific time interval like the early Holocene?
- Line 132-133: Throughout the manuscript and in this sentence the authors refer to "P-E" but here they are demonstrating that the "E" component is not significant. Why not just state that Lake Junín is especially sensitive to "precipitation amount" ? There are lots of other factors that control E other than temperature (e.g. wind strength) but those do not appear to be within the scope of the manuscript, so it would be simpler to just talk about P.
- Also related to lines 130-133 and in general: How fair is the interpretation of TOC during that time interval (and for the whole Holocene) vs. during the glacial regime in the catchment? Does the TOC proxy function in the same way when the catchment is glaciated vs. not? This should be discussed/clarified.
- Lines 150-153: Such an interesting observation here, it could be worthwhile to point out that DO event 13 is also weaker at NGRIP.
- Lines 169-171: The peat record shows these two prolonged droughts, but the Ti record does not show the same thing. Important to discuss the disagreement of these 2 proxies here since large claims about the mechanisms of deglaciation are being made.
- Lines 226-227: Include this statement about the meaning of asterisks in the caption for Figure 2 as well.
- Lines 259-260: *All* datasets generated during the current study should be made available on the NOAA website. Datasets published in Supplementary Material tables are notoriously difficult to locate and to machine-read.
- Lines 267-268: List the award numbers for ICDP and NSF that supported this work.

Reviewer #3:

Remarks to the Author:

This paleoclimatic study by Woods et al. presents original research results based on a sedimentary core drilled in the Junin lake, which is located in the High Tropical Andes in Peru. This record covers the last 50 ka. They analyzed several parameters in this core, notably the silico-clastic content and the total organic carbon. They interpreted these data as glaciers fluctuations driven by precipitation changes. They attribute these changes to AMOC-Greenland D/O oscillations, North Atlantic warm episodes (interstadial) being in phase with the shrinkage of tropical Andes glaciers, while glacial advances are in phase with cold Heinrich events. Authors attribute these Tropical glacier fluctuations to hydroclimatic oscillations driven by the AMOC. The main strength of this new sedimentary archive is its continuity and its high temporal resolution that permits comparison with other well-dated continuous paleoclimatic archives (Greenland ice, speleothems, oceanic sediments). This is an original dataset which brings useful complementary data to our knowledge of the link between high latitude and tropical regions over the last 50 ka. The existence of a link between the glaciers mass balance and the AMOC was already documented for the last deglaciation. Here, authors extend the reality of this mechanism to 50 ka. However, I noticed several issues and I thus raise several concerns that should be taken into account during the revisions of the manuscript.

Major concerns

1 - Important previous work is overlooked

Authors ignored the existence of several important articles that document the fluctuations of lake levels and glaciers in the Tropical Andes in link with the AMOC variability (Placzek et al., GSA Bull. 2006 ; Blard et al., QSR, 2011 ; Martin et al., Sci. Adv., 2018 ; Palacios et al., Earth Sci. Reviews, 2020). These studies already showed that glacier and precipitation fluctuations in the Tropical Andes are tightly paced by AMOC millennial abrupt changes that occurred since the Last Glacial Maximum.

By ignoring this scientific literature, authors don't describe properly the state of the art and they overlook the importance of lake level fluctuations as tropical paleo-precipitation proxies (Placzek et al., GSA Bull., 2006 ; Blard et al., QSR, 2011 ; Martin et al., Sci. Adv., 2018). Several sentences are hence overstated, suggesting that we know little, or even nothing, about the behavior of glaciers and the glacial-interglacial hydroclimatic changes that occurred in the Tropical Andes. These sections should be revised, by quoting the existing literature. For example:

Lines 45-46: "... little is known about DO-related precipitation anomalies."

Lines 47-48: "Much of the paleoclimatic evidence documenting changes in South American hydroclimatic changes relies on the interpretation of $\delta^{18}O$ variations in speleothems from the Amazin Basin...". Paleolake levels are also valuable paleoprecipitation proxies that brought useful insights about the hydroclimatic evolution of the Tropical Andes (Placzek et al., GSA Bull. 2006 ; Blard et al., QSR, 2011)."

Lines 65-66: "This record documents for the first time the unambiguous impact on glacier mass balance and hydroclimate of the climatic teleconnection linking the Atlantic meridional thermal gradient with the strength of the SASM". This is not correct. Several studies already reached this conclusion from well-dated paleolake levels and paleo-glacier extents (e.g. Martin et al., Sci. Adv., 2018; Placzek et al., EPSL, 2013).

2 - Comparison with existing records – update of the moraine ^{10}Be cosmic-ray exposure ages

Authors compare their core results with two studies providing ^{10}Be cosmic-ray exposure ages (Smith et al., Science, 2005 and Shakun et al., JQS, 2015). The ages of (Smith et al., Science, 2005) should be recomputed with the most recent regional production rates that were published in the Tropical

Andes (Martin et al., *Quat. Geoch.*, 2015; Kelly et al., *Quat. Geoch.*, 2015). This necessary recalculation may change by 500 to 1000 years the LGM ages published in (Smith et al., *Science*, 2005). This geochronological update is important before comparing their new record with paleo-glacier fluctuations.

3 – Use of the 14C calibration curve

A recent paper by Cheng, Edwards et al. (*Science*, 2018) proposed an updated 14C calibration curve from the Hulu Cave stalagmites that are well dated with the U/Th chronometer. This new curve improved the accuracy of the radiocarbon chronometer, notably for the 35-50 ka period. I strongly encourage authors to check the impact of this new up-to-date calibration curve on their age model.

Other issues

Line 32-34: Alternative scenarios may also occur. Future warming could on the contrary reduce the AMOC, leading to a precipitation increase in the Tropical Andes (cf papers by Broecker and Putnam).

Lines 105-106, Fig. 2 and 3 and elsewhere: I have been worried reading that Si and Ti concentrations are reported in cps (count per seconds). This unit is instrument-dependent and does not tell much about the real concentrations of these elements. Could you use a transfer function to provide these data in weight% or in any other unit that has a physical meaning?

Lines 101 to 122 and elsewhere: How can you be sure that silico-clastic maxima are really synchronous with glacier maxima? Can't we imagine that glacial recession periods yield a transient sediment discharge? This would lead to an alternative scenario. A rational justifying this reasoning would be useful.

Lines 131-132: This statement is a bit misleading. P-E (= lake level variations), are both dependent to precipitation (P) and temperature (temperature controls evaporation E). I suggest rephrasing these lines to highlight that lake level variations are mainly driven by precipitation changes.

Lines 164: In some regions, the Tropical Andes glaciers started to retreat well before 22 ka (e.g. Blard et al., *QR*, 2014). On the opposite, deglaciation occurred after 20 ka in other parts of the Tropical Andes (see Palacios et al., *ESR*, 2020).

Line 187: Without any modeling combining the lake budget and glacier mass balance (e.g. Martin et al., 2018), the design of your study does not permit to separate the respective impacts of precipitation and temperature on the glacier-lake changes that you observe.

Line 216: What is a "small sample mass" and a "low CO₂ gas yield"? Provide numbers.

Figure 1: It would be useful to add here a regional map of South America with the main atmospheric features.

Reviewers' comments:

Reviewer #1 (Remarks to the Author):

The manuscript by Woods et al. describes a high-resolution record of landscape history from Lago Junin in the tropical Andes over the last 50k years and its relationship to Dansgaard Oeschger variability, first identified in the Greenland ice core data. The data are of very high quality, and the major results confirm prior studies on millennial variability in tropical South America in documenting the waxing and waning of Andean glaciers over this period in response to variation in precipitation driven by the South American Summer Monsoon (SASM). The results are similar to those of the Lake Titicaca record (Fritz et al. 2010), which is not as well dated as the Lago Junin record presented in this paper and does not have the high temporal resolution provided by scanning XRF. Yet, it clearly shows millennial (D-O) variation in the period between 60 and 20 ka, and this variability was attributed to differential input of clastic sediments, because of variation in precipitation (SASM) and variation in clastic material from precipitation sensitive Andean glacier advance and retreat. In addition, several speleothem records from the eastern slope of the Andes document SASM variation associated with D-O variability with high temporal precision, as acknowledged in this manuscript. The submitted manuscript in its current form overstates the novelty of the results and has some limitations in scope and interpretation that I summarize below.

We recognize that prior studies have documented millennial-scale precipitation variability in tropical South America (lines 40-44, 157-161). Some lacustrine records have postulated a link between DO events and Andean climate on the basis of this millennial variability, including the work from Lake Titicaca. However, these records lack the temporal resolution and chronologic precision to link individual DO events to Greenland ice cores or well-dated speleothems.

The reviewer's assertion that the drill cores from Lake Titicaca "clearly shows millennial (D-O) variation in the period between 60 and 20 ka" is unfounded. The correspondence between the 33 wet intervals identified in the Lake Titicaca record and the 16 cold intervals of the NGRIP record is tenuous and suggests substantial variability that cannot be explained by a connection with the North Atlantic. Further, the interval spanning 20-41 ka contains only 12 bulk sediment radiocarbon ages, and the age model from 30-60 ka is based solely on linear extrapolation from an age estimate of ~30 ka at 15.69 mblf to the interval identified as the onset of the last glacial cycle at 42 mblf (Fritz et al, 2007). The resultant age (~65 ka) is at odds with 4 U/Th ages only 3.8 m deeper in the core that yield an average U/Th age of 122.8 ka (Fritz et al 2007). With only 5-10 samples per meter, the discontinuous proxy measurements of the Titicaca record limit one's ability to speculate on the identification of individual D-O events, and the age model precludes confident correlation with the NGRIP record. In contrast, the Junin record is based on 75 terrestrial macrofossil radiocarbon ages (44 ages in the 20-50 ka interval) anchoring a series of proxy records sampled at mm- to cm-scale.

Additionally, Lake Titicaca cannot be considered a record of Andean glacier advance and retreat during the last glacial period. As noted by Fritz et al (2010) no moraines have been identified in the Titicaca drainage basin that date to the interval 30-60 ka and all moraines in the Titicaca basin are much more distant to the lake compared to Junin, where moraines reach

to within 1-2 km of the lake shore. While a DO connection in the Lake Titicaca record was hypothesized based on millennial changes in biogenic silica, benthic diatoms, grain size, and $\delta^{13}\text{C}$ of organic matter, these proxies are said to fluctuate in response to fluvial discharge and input of allochthonous material of nearshore origin (Fritz et al, 2010), but are not themselves diagnostic of glacially generated clastic material. In short, the glacial connection in the Junín record is fundamental to the interpretation of the Junín cores; this is not true of the Titicaca cores that are located >50 km from shore in 230 m of water – it is much more likely that the medium to coarse silt came from a local shoreline during water level changes and is unrelated to glaciation. We have revised the manuscript to explicitly mention the millennial scale fluctuations in terrigenous inputs to Lake Titicaca and their possible connection to DO events (lines 153-155).

Woods and colleagues suggest that there are ambiguities in speleothem records of hydroclimate in terms of the relationship between local precipitation amount and ^{18}O , necessitating additional proxies, such as lake records (line 51-57), to validate interpretations. It is true that ^{18}O records are influenced by multiple variables and hence are not always unambiguous recorders of precipitation amount. Yet the two studies that they cite from South America (Wortham, Ward, references #11,12) show coherent signals in ^{18}O records from sites within the heart of the SASM, such as the Andean highlands and slopes, and do not suggest that alternative interpretations of the original publications are warranted for speleothem ^{18}O results from this region of tropical South America. Thus, both cited studies interpret the speleothem records from the Andes and western Amazon as effective recorders of changes in SASM intensity and local precipitation patterns. The sites where ^{18}O and Sr/Ca reconstructions show some divergence, and where the Wortham and Ward papers postulate that moisture sources in addition to monsoon-sourced precipitation might be important, are all in the eastern Amazon and southeastern Brazil, not the Andes. These regions have somewhat more complex precipitation influences (e.g. from the SACZ). It is also noteworthy that these studies are Holocene in age, a period when the SASM monsoon was likely not as strong as during the cold stadials of MIS2 and MIS3.

We respectfully disagree with the reviewer's characterization of the main findings of the two studies we cite (Wortham et al., 2019, Ward et al., 2017) regarding ^{18}O interpretation in South America. We emphasize that these studies demonstrated that South American speleothems cannot be assumed to be effective recorders of local precipitation patterns in the absence of confirmation from additional proxies. Wortham et al 2017 shows a coupling between several speleothem $\delta^{18}\text{O}$ records of regional monsoon intensity spanning the past 2 ka and local moisture availability inferred from the Sr isotope record of their site in central Brazil – but that $\delta^{18}\text{O}$ records from central and southeast Brazil are decoupled from local moisture and from regional monsoon intensity. This study thus demonstrates divergent $\delta^{18}\text{O}$ patterns within and outside the core monsoon region of the Amazon Basin, and highlights the need for paired high resolution isotopic and non-isotopic records.

Similarly, Ward et al 2019 found that during the Holocene, speleothems from peripheral monsoon sites (e.g. southeast Brazil and the Peruvian Andes, including the location of Junín) follow insolation, whereas those from the interior convective region (north-central, south-central, and eastern Brazil) do not, suggesting they are not sensitive to local insolation changes

and the associated ITCZ shift during the Holocene, or that there are additional influences on speleothem $\delta^{18}\text{O}$. The lack of consistency between Sr isotopes and $\delta^{18}\text{O}$ observed in speleothems from multiple sites suggests they may not be a reliable proxy of local moisture conditions, or that for some sites the local response to regional monsoon intensity is not consistent through time. These findings further demonstrate that a single isotope record alone can lead to erroneous interpretations about local moisture conditions, and they underscore the need for in-situ corroboration from other precipitation-sensitive proxies. The Junín record paired with the Pacupahuain speleothem fills this gap for the 17-50 ka time period in the high-elevation tropical Andes.

It is also true that clastic flux and the associated geochemical signals in a sediment core are influenced by multiple variables, and thus these are not unambiguous recorders of precipitation or of glacial mass balance as suggested (line 66). The authors treat the Si and Ti signals as semi-quantitative records of precipitation (lines 150-159), when they are not; they are qualitative signals of wet versus dry. Thus, there is no reason that the magnitude of ^{18}O variation in the nearby Pacupahuain speleothem record and the clastic signal in Lake Junin should scale equally in response to the same precipitation forcing. The assumption that the the Lago Junin record is the real measure of local precipitation (lines 157-162) is not well founded, and the associated inference that the speleothem records are more ambiguous recorders of local precipitation amount than the record from Lago Junin is not supported.

While it is true that proxy indicators of glacial sediment input (clastic flux, bulk density, Si and Ti, among others) and of lake level lowering (high TOC and presence of peat) can be influenced by other variables, we contend that for the Junín watershed, these parameters are highly diagnostic of the presence or absence of glaciers in the watershed and of substantially lowered lake levels, respectively. The foundational work reported in Seltzer et al. (2000) and Smith et al (2005) demonstrate the sedimentological linkage between glaciation and sedimentology in Lake Junin. In addition, the *duration* of dry events are legitimately comparable on a quantitative basis. Further, speleothems in this region are well known to primarily record the amount of rainfall upstream over the Amazon Basin (e.g. Vuille et al, 2012), with local impacts being less well established. Nonetheless, the reviewer is correct that we do need to avoid over-quantifying our interpretation of the proxy data when comparing the Pacupahuain and Junín records, and we have revised our text accordingly (lines 168-170).

The paper makes no comparison of the Lago Junin record to the nearby Lake Titicaca record (Fritz et al. 2010), despite the many similarities between the two records and the similar focus of the earlier publication on Lake Titicaca. There are clear advantages to the Lago Junin record, such as the greatly improved age control throughout and the highly resolved record. Yet the paper does not take the opportunity to build on prior foundational studies or to acknowledge the nearby Lake Titicaca record of water level and climate change associated with DO events (e.g. lines 136-140).

The reviewer is correct here and in the revised manuscript we explicitly mention the millennial fluctuations observed in the Titicaca record (lines 153-155) and we expand our discussion to include other Altiplano records as well (lines 48-50, 151-161).

It is noteworthy that the paper says nothing about Heinrich events, despite considerable evidence in tropical South America from speleothems, lake cores, and marine cores that these were among the largest excursions in the hydroclimate of tropical South America during MIS3-2. The absence of a clear signal of the Heinrich events in Lago Junin warrants discussion given the temporal focus of this paper.

As pointed out by the reviewer, it is notable that the Junín proxy data reveals that the signal of Heinrich events is not as striking as the signal of DO events. We added some discussion of Heinrich events to the revised manuscript (lines 48-50, 163-167). However, we note that the majority of Heinrich events occur during wetter periods when there is already increased glacial sediment input making them harder to detect as they get masked by already high values of Ti and Si, and that Heinrich Event H1 occurred after complete deglaciation of the Junín catchment (Smith et al., 2005), so there is no glacial sediment input during this interval. In addition, the Junín record may be more biased toward recording dry, interstadial (DO) events than wet, stadial (Heinrich events) because the addition of more ice to an already glaciated catchment does not have the same sedimentologic impact as does the removal of ice from the catchment. In essence the clastic signal becomes saturated and is not as sensitive to additional ice volume, whereas ice retreat behind moraine dammed lakes can result in a profound reduction in clastic input, and this was clearly noted by Seltzer et al (2000). The lake level proxies also exhibit a bias towards moisture deficits – peat layers and high TOC content signal lowstands, but we lack an equivalent signal for high lake level because once the lake begins to overflow its response higher precipitation is limited.

We also note that Heinrich events were not interpreted in the Titicaca record (Fritz et al. 2010), and are not nearly as marked as are DO events in the Pacupahuian $\delta^{18}\text{O}$ record (Kanner et al 2012).

It is also interesting that available records indicate the climatic signature of Heinrich events varies regionally in South America, and many records are characterized by inconsistent responses during Heinrich events, with possible differences among the marine and terrestrial realms. As summarized in Lynch-Stieglitz (2017), H4 and H5 show a clear weakening of AMOC in several marine records, whereas the changes are muted or absent during H2 and H3, implying a stronger response of AMOC during the milder MIS 3 interval compared to MIS 2. A recent study (Zhang et al, 2017) looked at terrigenous inputs in marine cores from the western equatorial Atlantic off the northeast coast of South America, and found that sites in the northern tropics showed variations of similar magnitude among Heinrich and DO stadials, whereas sites near and just south of the equator showed large terrigenous inputs during Heinrich events but minimal variability associated with DO events, implying a possible latitude-dependent response. Finally, speleothems from both the western Amazon (El Condor/Diamante, Cheng et al, 2013) and eastern Amazon (Paraiso Cave, Wang et al, 2017) display variations in SASM activity that are of similar magnitude during Heinrich events and DO stadials, and the Pacupahuian speleothem is notable for its anomalous responses during some

Heinrich events, as discussed in Kanner et al, 2012. In this record, both H4 and H5 are interrupted by positive isotope excursions indicative of an abrupt weakening of the SASM. A similar behavior is observed in the Junín Ti and Si data for these two events, suggesting precipitation anomalies that were specific to this region of the Andes.

A last paragraph that emphasizes the new insights from the Lago Junin record itself would improve the manuscript. Multiple prior records and publications have documented the sensitivity of tropical P-E balance to Northern Hemisphere climate perturbations. Similarly, there are a number of detailed modeling studies about the contemporary mass balance of glaciers in the tropical Andes and their potential response to 21st century warming that provide more insight about the potential future fate of regional glaciers and water supply.

We have expanded the discussion in the last paragraph to include additional scenarios of future tropical hydrologic change and to acknowledge the contrast in the expected future AMOC response compared to the last glacial period.

In summary, I think a reframing of various aspects of the manuscript, as suggested above, would help to showcase the high-quality results and generate a more effective addition to the literature on Andean paleoclimate variability.

Sheri Fritz, University of Nebraska - Lincoln

Reviewer #2 (Remarks to the Author):

This manuscript by Woods et al. presents a truly fascinating and unique study on glacial and deglacial climate in the tropical Andes, using continuous and high resolution sediments from Lake Junín drill cores. The paper argues for a connection between warm interstadials in Greenland and moisture balance, as well as glacier mass balance, in the tropical Andes. The mechanism is that DO warm events in Greenland correspond to higher TOC and the formation of peat at/near the core location, along with a decrease in siliciclastic flux, all of which indicate dry intervals in the Junín watershed. Due to the sensitivity of glaciers in this part of the Andean tropics to precipitation (rather than just temperature), the authors further argue that these dry periods also correspond to decreases in local glacier mass balance. These results yields two high impact implications: First, that the amount of precipitation in the Junín catchment is indeed tied to South American Monsoon variability, just as studies based on the d18O of precipitation (ice cores, speleothems) have found, but only generally speaking; the relationship is not 1:1 because of local factors that may enhance or decrease precipitation in the absence of corresponding variations in the SASM. So, variations in d18O should not necessarily be interpreted as a % increase/decrease in local precipitation amount although the overall relationship holds. Second, this paper presents a new mechanism for the “early deglaciation” previously observed in the tropical Andes: That precipitation-sensitive glaciers were first diminished by drought, and then were further diminished by deglacial warming.

The paper is novel and exciting, and presents some really special and new food for thought for the paleoclimate community. It is certainly worthy of publication in Nature Communications. However, there are some important issues that need to be resolved before the manuscript can merit publication. The major issues are outlined below, followed by minor issues.

Major comments:

1) Age model. The number of radiocarbon dates for this record is truly impressive so I hate to provide negative feedback here, however there is a large gap in ^{14}C dates submitted between 22,600 kyr BP and 28,900 kyr BP. This is problematic because it questions the evidence for the following claims that the authors make:

With 79 AMS radiocarbon ages spanning the past 50 kyr, this may be the best dated sedimentary record of this interval from the tropical Andes. Nonetheless, the reviewer is correct in noting that the interval from 22.5 to 28.9 ka is not constrained by radiocarbon ages. This is because this interval is the LGM in the tropical Andes, glaciers were generally in advanced positions during this interval and large amounts of glacial flour—devoid of organic matter—were introduced to the lake. The sediment, with few exceptions, is all glacial flour, high in Ti and Si, and low in TOC. Simply put, after repeated attempts we found nothing to date in this section of the core. We have amended the text to clarify this point (lines 113-114, 139-141).

- “Glaciogenic sediment input to Lake Junín was especially high from 28.5-22.5 ka, which corresponds to the age of moraines deposited during the maximum extent of ice in the last 50 ka in the adjacent eastern cordillera” (lines 106-109). The timing of the high glaciogenic input is questionable here. All that is clear is that it was high at some point before 22.5 ka and after 28.5 ka. But it is quite possible that sedimentation rates proceeded at their pre-28.5 ka values until 23 ka or 24 ka or any other time within that interval. Assuming that the sed rate changed essentially right after the last ^{14}C date prior to the interval seems unfair.

While we cannot know precisely when the sedimentation rates changed across this interval, we do see that Ti and Si increased abruptly at the beginning of this interval, consistent with accelerated clastic input. Furthermore, the age modelling software that we used in this study (Bacon), does not simply linearly interpolate between dated intervals as other age modeling software packages do. A defining feature of Bacon is that it uses a gamma autoregressive process to model accumulation rates based on prior information to establish a coherent evolution of deposition along the core, whereby the software integrates a “memory” of other dated intervals to allow for a more realistic shape or smoothness of the accumulation rates (Blaauw and Christen, 2010).

- “There is no evidence... for unconformities” (lines 115-116). In fact there is either a very

large sedimentation rate change that just so happens to correspond to 28.5 ka (as suggested in the lines referenced above) or there are hiatuses, or both. It seems extremely unlikely that an 8m deep lake would come close enough to drying out that it leaves behind peat deposits, but manages somehow to not dry out completely.

Respectfully, the reviewer has this backwards. What we see is an increase in sed rates beginning sometime around 28.5 ka and extending to 22.5 ka. A hiatus would be a decrease in overall sedimentation rates. To be sure, without age control over this interval it is possible that there were short hiatuses during this interval of otherwise high sedimentation rates. But the sediment record suggests that this likely did not happen because of the uniformly high indicators of glacial flour input, which would require wet conditions and high lake level, and the general absence of peat layers during this interval. During other intervals of the record where peat was deposited we cannot rule out short hiatuses, but the dense radiocarbon data set clearly demonstrates that any such hiatuses were short and had little to no effect on the age model.

We do recognize, however, the paradox of a lake that is today 8-12 m deep repeatedly fluctuating but not drying out in the past. We note some clarifications here, and in the text (lines 119-126), to address this topic. First, it's likely that the average lake level lowered by several meters, allowing for wetlands to expand towards the lake center, but the very seasonal nature of monsoon precipitation in this region likely continued to submerge the depocenter during the summer wet season. This is not surprising given that all surface water draining from this large, high elevation watershed must pass through the Junín lake basin, making it nearly impossible to dry out completely. Additionally, while the lake is relatively shallow, its large surface area (300 km²) and the cold temperatures at high elevation during the last glacial period likely suppressed evaporation, making it less prone to complete desiccation. Small contributions of glacial meltwater may have buffered lake volume during the dry season as well. Second, we cannot rule out the possibility that the lake dried out entirely for multiple consecutive years at a time, and while the stratigraphy does not reveal any obvious unconformities, we can only say that based on the density of radiocarbon ages, physical measurements of sediment characteristics, and core sedimentology, there is no evidence of significant hiatuses.

- "The Junín record exhibits a reduced input of glaciogenic sediment during DO interstadials 3-13" (line 124). DO interstadial 3 is suspect because of the lack of age control in the Junín record at this time.

The reviewer is correct that the age control around the time of DO 3 is less robust and we have amended the text (lines 138-141) to acknowledge the uncertainty associated with the timing of DO events 2-3.

It seems that the gap in age control points is between cores D15-D9 and 1B-4H-1. If there is some reason for this (is this interval devoid of macrofossils?) then the authors need to be very clear about that and need to explain how the lack of age control in this important interval might complicate their interpretations of climate during the LLGM.

As noted above, this gap in age control points corresponds to a nearly homogenous deposit of fine-grained glacial clay that is devoid of macrofossils or other organic material suitable for ¹⁴C dating, and it is not a feature of a gap between cores. We have clarified this on lines 113-114 and 139-141 of the revised manuscript.

2) Overall relationships between glaciogenic influx and lake level. Clearly the glacial period and the Holocene are two very distinct regimes in this lake, with the glacial period being characterized by high inputs of Ti, Si, and siliciclastic flux but a very different regime once the catchment was fully deglaciated. The reason that lake levels were high during the glacial period is not the same reason they are high today (which is actually not clear—why are they so high today anyway??) If a non-paleolimnologist were to read this paper there would be confusion about why you cannot look at high Ti during the glacial period and infer high lake levels, but then look at low Ti during the Holocene and infer low lake levels (since the lake level today is quite similar to the highest highstands during the glacial period). To this end I suggest:

- Devote some text in the introduction to a clear description of the physical processes by which higher ice volume leads to higher glaciogenic sediment/siliclastic flux. In addition to the confusion mentioned above it is a little counter-intuitive that one would not associate high glaciogenic flux with warm intervals, for example (and subsequent melting of glaciers in the catchment leading to higher runoff to the lake). I can assume some of the physical reasoning behind this but it needs to be explicitly stated in the manuscript. If there are substantial lead or lag times between glacier advance and glaciogenic input this needs to be stated and discussed too.

The Junín catchment was completely deglaciated by about 15 ka, and this is supported by cosmogenic radionuclide dating of polished bedrock cirque headwalls in one of the principal glaciated valleys to the east of Lake Junín (Smith et al., 2005). This was further confirmed by Seltzer et al (2000) as the timing of clastic sediment cessation to Junín. There are no late glacial or Holocene moraines in any catchment surrounding the lake. The Ti, Si, and clastic flux records reported here are entirely consistent with this chronology. We added additional text (lines 100-105, 128-134) to emphasize that the very low levels of Ti, Si, bulk density, and siliciclastic flux after 20 ka and throughout the Holocene indicate the retreat of glaciers behind moraine-dammed lakes and associated loss of the clastic signal, and added more detail describing the differences between the glacial and Holocene sediments.

Apparently, the events that caused the peat layers during MIS 2-3 ceased during the late glacial and Holocene. Given that these events are causally associated with DO events in the Greenland ice cores, and since DO events also did not continue into the late glacial and Holocene in Greenland, this is consistent with our interpretation of the Junín record. We do not, however, take the absence of peat layers during the Holocene to mean that lake levels were consistently high, but rather that the sedimentary regime had changed. TOC during this time was governed more by the deposition of organic rich lake sediments likely reflecting in-lake productivity which interrupt the generally high carbonate content of the lake mud.

The complete and final deglaciation of the Junín catchment WAS associated with late glacial warming that is documented in other regional records (e.g., Huascarán ice core—Thompson et

al., 1995) and in regional snow line reconstructions. This warmth is what is different about the terminal Pleistocene deglaciation of Junín compared to the brief and abrupt deglacial cycles that correspond with DO events. During these latter events, it was ice starvation by reduced precipitation rather than warming that drove the signal. In contrast, the warming that drove glaciers out of the Junín catchment by 15 ka was not associated with especially dry climates, and even during the insolation minima at 9 ka, there is no evidence of widespread lake desiccation.

In terms of melting ice and increased run off to the lakes and resultant 'paraglacial' sedimentation (Church, 1972), this, must have occurred but we do not have the resolution in our record to delineate *initiation of glacier retreat* from *meltwater pulse* from *final lake lowering*. It is likely that these events happened very rapidly, perhaps on the scale of regional ice retreat today. What this also implies is that the arid phases that were driven by DO events in the Greenland sector were strong enough in the western Amazon basin to drive down regional lake level in spite of melting ice. Also, any paraglacial sediment pulse may have been captured by the paternoster, moraine-dammed lakes that formed upon initial ice retreat. This causes the deglacial sediment record to be non-linear, and highly sensitive to initial ice retreat.

- Consider adding some notation to Figure 2 that shades or otherwise denotes the deglaciated portion of the record.

We prefer not to crowd Figure 2 with additional notations, but the inclusion of an X axis with tick marks on top of the figure (as requested below) makes it easier to see that the final pulses of Ti and Si, and thus deglaciation, corresponds to ~18 ka, as noted in the text.

- Consider adding TIC data to Figure 2 since the text says that carbonate increases substantially after deglaciation and, presumably, one would interpret variations in TIC vs. TOC vs. siliciclastic flux quite differently.

The reviewer is correct that the TIC data and its context warrant inclusion in the manuscript. We added a Supplemental Figure 2 that includes TIC (CaCO₃ wt. % and scanning XRF counts of Ca), dry bulk density and siliciclastic flux, and we describe the sedimentological interpretation of these proxies before and after deglaciation on lines 464-470. We leave this discussion out of the main text due to space limitations.

We also note that the siliciclastic flux calculation removes both TOC and TIC so that variations in siliciclastic flux are not dependent on these two components (lines 269-278).

3) Figure clarity: The timing of events in the records are rather hard to read because of the tall orientation of Figures 2 and 3 and the small tickmarks on the X-axis. Two suggestions:

- Repeat the X axis on top of the figure so that it is easier for readers to figure out the timing of events in the different timeseries
- Move figure 2f to a separate panel within the figure, giving Figure 2a-e its own set of X axes (top and bottom). Then, add triangles to the top X axis denoting where age control points occur (as is common with speleothem papers)

- In Figure 3, shade or otherwise denote the time periods toward the end of the glacial period where the peat deposits occur (roughly 22.5-20.1 ka) and where the “dry intervals cause early onset of glacial retreat” argument is rooted (lines 170-172)

We appreciate these suggestions to improve figure clarity, and adding an upper X axis has improved the readability for both Figures 2 and 3, but we prefer to leave Figure 2f where it is rather than in a separate panel.

Minor comments:

- Line 39: Change this to “South American low-latitude paleoclimate proxy records” since this sentence talks about the SASM and not low latitude records in general

Done.

- Lines 75-78: How do we know that “at no time during at least the last 50 ka has the lake been overridden by glacial ice” ? Is there no evidence for moraines along the southwestern shore in Fig 1?

The mapped ice limits from Smith et al (2004) preclude ice covering the lake basin at any time in the past 50 ka (though actually much longer than that), and there are no moraines on the southwest side of the lake.

- Lines 85-87: Provide a citation for the “less than 7% falling during the winter (JJA)” number

This number is based on unpublished rainfall data (1916-1998 CE) for La Oroya, Peru, from the Servicio Nacional de Meteorología e Hidrología del Perú, Ministeria del Ambiente. It appears from the Nature Communications submission guidelines that data sets can only be cited in the references if they have a DOI, and footnotes are not used, so we defer to the Editor on whether we can cite this information.

- Throughout manuscript: Shouldn't the term be spelled glaciogenic, not glaciagenic?

Both spellings are in use in the literature.

- Lines 130-133: This statement is not very clear. What time period exactly are they referring to as the “late glacial-to-Holocene transition” ? It is confusing because TOC does increase during the deglaciation, unlike what they are saying here, but perhaps they are referring to some specific time interval like the early Holocene?

We revised this sentence for clarity (lines 145-149).

- Line 132-133: Throughout the manuscript and in this sentence the authors refer to “P-E” but here they are demonstrating that the “E” component is not significant. Why not just state that Lake Junín is especially sensitive to “precipitation amount” ? There are lots of other factors that control E other than temperature (e.g. wind strength) but those do not appear to be within the scope of the manuscript, so it would be simpler to just talk about P.

While Junín appears more sensitive to precipitation amount, we cannot rule out changes in E so it is probably still more accurate to state this as P-E.

- Also related to lines 130-133 and in general: How fair is the interpretation of TOC during that time interval (and for the whole Holocene) vs. during the glacial regime in the catchment? Does the TOC proxy function in the same way when the catchment is glaciated vs. not? This should be discussed/clarified.

As noted above:

TOC during the Holocene was governed more by the deposition of organic rich lake sediments likely reflecting in-lake (algal) productivity, and is indicative of a different sedimentary environment than the glacial-age peat layers. We have clarified this in lines 130-134 of the revised manuscript.

- Lines 150-153: Such an interesting observation here, it could be worthwhile to point out that DO event 13 is also weaker at NGRIP.

- Lines 169-171: The peat record shows these two prolonged droughts, but the Ti record does not show the same thing. Important to discuss the disagreement of these 2 proxies here since large claims about the mechanisms of deglaciation are being made.

We are uncertain what the reviewer is referring to here – the Ti record, as well as the Si, siliclastic flux, and bulk density all show clear declines in glacial inputs from 22.5-21.9 and 20.8-20.1 ka that coincide with the peat layers associated with these deglacial drought episodes.

- Lines 226-227: Include this statement about the meaning of asterisks in the caption for Figure 2 as well.

Done.

- Lines 259-260: *All* datasets generated during the current study should be made available on the NOAA website. Datasets published in Supplementary Material tables are notoriously difficult to locate and to machine-read.

We will make all datasets available on the NOAA website including those in the supplementary material.

- Lines 267-268: List the award numbers for ICDP and NSF that supported this work.

Done.

Reviewer #3 (Remarks to the Author):

This paleoclimatic study by Woods et al. presents original research results based on a sedimentary core drilled in the Junin lake, which is located in the High Tropical Andes in Peru. This record covers the last 50 ka. They analyzed several parameters in this core, notably the silico-clastic content and the total organic carbon. They interpreted these data as glaciers fluctuations driven by precipitation changes. They attribute these changes to AMOC-Greenland D/O oscillations, North Atlantic warm episodes (interstadial) being in phase with the shrinkage of tropical Andes glaciers, while glacial advances are in phase with cold Heinrich events. Authors attribute these Tropical glacier fluctuations to hydroclimatic oscillations driven by the AMOC. The main strength of this new sedimentary archive is its continuity and its high temporal resolution that permits comparison with other well-dated continuous paleoclimatic archives (Greenland ice, speleothems, oceanic sediments). This is an original dataset which brings useful complementary data to our knowledge of the link between high latitude and tropical regions over the last 50 ka. The existence of a link between the glaciers mass balance and the AMOC was already documented for the last deglaciation. Here, authors extend the reality of this mechanism to 50 ka. However, I noticed several issues and I thus raise several concerns that should be taken into account during the revisions of the manuscript.

Major concerns

1 - Important previous work is overlooked

Authors ignored the existence of several important articles that document the fluctuations of lake levels and glaciers in the Tropical Andes in link with the AMOC variability (Placzek et al., GSA Bull. 2006 ; Blard et al., QSR, 2011 ; Martin et al., Sci. Adv., 2018 ; Palacios et al., Earth Sci. Reviews, 2020). These studies already showed that glacier and precipitation fluctuations in the Tropical Andes are tightly paced by AMOC millennial abrupt changes that occurred since the Last Glacial Maximum.

By ignoring this scientific literature, authors don't describe properly the state of the art and they overlook the importance of lake level fluctuations as tropical paleo-precipitation proxies (Placzek et al., GSA Bull., 2006 ; Blard et al., QSR, 2011 ; Martin et al., Sci. Adv., 2018). Several sentences are hence overstated, suggesting that we know little, or even nothing, about the behavior of glaciers and the glacial-interglacial hydroclimatic changes that occurred in the Tropical Andes. These sections should be revised, by quoting the existing literature. For

example:

Lines 45-46: "... little is known about DO-related precipitation anomalies."

Lines 47-48: "Much of the paleoclimatic evidence documenting changes in South American hydroclimatic changes relies on the interpretation of $\delta^{18}O$ variations in speleothems from the Amazon Basin...". Paleolake levels are also valuable paleoprecipitation proxies that brought useful insights about the hydroclimatic evolution of the Tropical Andes (Placzek et al., GSA Bull. 2006 ; Blard et al., QSR, 2011)."

Lines 65-66: "This record documents for the first time the unambiguous impact on glacier mass balance and hydroclimate of the climatic teleconnection linking the Atlantic meridional thermal gradient with the strength of the SASM". This is not correct. Several studies already reached this conclusion from well-dated paleolake levels and paleo-glacier extents (e.g. Martin et al., Sci. Adv., 2018; Placzek et al., EPSL, 2013).

Our revised text provides a more complete review of the state of knowledge of late Quaternary hydroclimatic fluctuations in the tropical Andes (lines 48-52, 151-161). We discuss the record from Titicaca (lines 153-155) and provide (above) an expanded discussion of that data set. In addition, we do note the apparent correlation of altiplano lake high stands with the Younger Dryas and Heinrich Event 1 (Baker et al., 2001; Placzek et al., 2006, Blard et al., 2011), which clearly point to the position of the ITCZ and AMOC. We also note the moraine evidence of Martin et al (2018) which concludes that the position of the Bolivian High is also involved in the advection of Atlantic moisture to the altiplano.

While the altiplano records, which include the Titicaca and Uyuni drill cores, and shorelines and moraine chronologies, are foundational in the concept of AMOC driving tropical Andean hydroclimate during several of the Heinrich events, none of these records have the combination of precise age control and a continuity of record that enables precise correlation with the full series of rapid millennial DO events recorded in the Greenland ice cores over the past 50 kyr. The closest comparison in this regard is the Salar de Uyuni drill core (Baker et al., 2001), which spans the past 50 kyr but is based on only 12 finite AMS radiocarbon ages on bulk sediment, none of which are older than 41.8 kyr. It is interesting to note that the altiplano records all seem to reveal the strongest signal of wet events, which are tied to Heinrich and Younger Dryas coolings in the North Atlantic. The altiplano records do not reveal a strong signal of DO cycles, which are intervals of reduced Atlantic moisture advection to the tropical Andes. As we discuss in our revised text (lines 157-161), the records seem to be biased to record hydroclimatic conditions that contrast the most with their respective background states. Thus, while the dry altiplano records strongly record wet (Heinrich) events, the wetter western Amazonian records, such as Junín and Pacupahuain, most strongly record dry (DO) intervals.

2 – Comparison with existing records – update of the moraine ^{10}Be cosmic-ray exposure ages

Authors compare their core results with two studies providing ^{10}Be cosmic-ray exposure ages (Smith et al., Science, 2005 and Shakun et al., JQS, 2015). The ages of (Smith et al., Science, 2005) should be recomputed with the most recent regional production rates that were

published in the Tropical Andes (Martin et al., Quat. Geoch., 2015; Kelly et al., Quat. Geoch., 2015). This necessary recalculation may change by 500 to 1000 years the LGM ages published in (Smith et al., Science, 2005). This geochronological update is important before comparing their new record with paleo-glacier fluctuations.

Our comparison is based on recalculated ^{10}Be ages (Heyman et al., 2016) of the original Smith et al. (2004) data set. These ages thus are based on updated production rates and scaling factors. We note the use of the recalculated age (line 116) and include the Heyman et al (2016) citation.

3 – Use of the ^{14}C calibration curve

A recent paper by Cheng, Edwards et al. (Science, 2018) proposed an updated ^{14}C calibration curve from the Hulu Cave stalagmites that are well dated with the U/Th chronometer. This new curve improved the accuracy of the radiocarbon chronometer, notably for the 35-50 ka period. I strongly encourage authors to check the impact of this new up-to-date calibration curve on their age model.

The record of variability in atmospheric ^{14}C content from Hulu cave does not itself serve as a standalone calibration curve, which typically incorporate data from a variety of different archives to offset any systematic or regional biases associated with the individual constituent records. Our understanding is that the dataset of Cheng et al 2018 will be featured prominently the forthcoming IntCal20 calibration curve, and we will certainly assess the impact on our age model when that work is released.

Other issues

Line 32-34: Alternative scenarios may also occur. Future warming could on the contrary reduce the AMOC, leading to a precipitation increase in the Tropical Andes (cf papers by Broecker and Putnam).

There is a clear scenario for a northward shift in the thermal equator as the Northern Hemisphere warms more than the Southern Hemisphere due to the higher percentage of land mass amplified by differential reduction of sea ice coverage in the Arctic and Antarctic, as described in Broecker and Putnam (2013, PNAS). If the hydrologic response of tropical South America operates similarly in the future as it did during abrupt warmings in the Arctic/North Atlantic region during the last glacial period, it is reasonable to expect drying in the Junín region. As pointed out by the reviewer, the interhemispheric response observed in the paleoclimatic record is an imperfect analog for future change for several reasons, one being the possibility that CO_2 -induced warming may weaken the AMOC. However, as discussed in Broecker and Putnam (2013), “any cooling that may result from weakened North Atlantic overturning will be countered by the warming and reduction in sea ice due to radiative effects of increased atmospheric CO_2 concentrations.” So, while a reduction in AMOC may occur under future warming, it has not, to our knowledge, been suggested that a reduction in AMOC alone

would be a sufficient forcing to shift the thermal equator southward and lead to a precipitation increase in the tropical Andes.

Nevertheless, the reviewer's point is taken, and we agree that a discussion of alternative scenarios would improve the manuscript. We have added additional text and references to acknowledge the possible future reduction in AMOC as well as the scenario of enhanced tropical precipitation suggested by some models (lines 215-222).

Lines 105-106, Fig. 2 and 3 and elsewhere: I have been worried reading that Si and Ti concentrations are reported in cps (count per seconds). This unit is instrument-dependent and does not tell much about the real concentrations of these elements. Could you use a transfer function to provide these data in weight% or in any other unit that has a physical meaning?

While we recognize that cps data is not equivalent to concentration due to the need of calibration, we are looking at relative changes downcore; the absolute concentration of Si and Ti is not important to our record. We have plotted ratios of XRF data, such as Ti/Al, so that the values are dimensionless and we see similar trends to those plotted. We do clarify on lines 168-170 of the revised manuscript that these proxies represent relative rather than absolute changes in precipitation and glacier mass balance.

Lines 101 to 122 and elsewhere: How can you be sure that silico-clastic maxima are really synchronous with glacier maxima? Can't we imagine that glacial recession periods yield a transient sediment discharge? This would lead to an alternative scenario. A rational justifying this reasoning would be useful.

As noted above:

Any paraglacial sediment pulse may have been captured by the paternoster, moraine-dammed lakes that formed upon initial ice retreat. This causes the deglacial sediment record to be non-linear, and highly sensitive to initial ice retreat. We know that this occurred during the last deglacial cycle wherein initial ice retreat behind LGM moraine dammed lakes, resulted in an abrupt reduction in clastic sediment to Junín (Seltzer et al., 2000). Thus, the "paraglacial" sediment phase (Church, 1972), appears to have been very short in the Junín sediment record.

Lines 131-132: This statement is a bit misleading. P-E (= lake level variations), are both dependent to precipitation (P) and temperature (temperature controls evaporation E). I suggest rephrasing these lines to highlight that lake level variations are mainly driven by precipitation changes.

As noted above, while Junín appears more sensitive to precipitation amount, we cannot rule out changes in E so it is probably still more accurate to state this as P-E.

Lines 164: In some regions, the Tropical Andes glaciers started to retreat well before 22 ka (e.g. Blard et al., QR, 2014). On the opposite, deglaciation occurred after 20 ka in other parts of the Tropical Andes (see Palacios et al., ESR, 2020).

The timing of deglaciation indeed varies throughout the tropical Andes and we have clarified this statement to refer specifically to the central Peruvian Andes in the modified text (line 187).

Line 187: Without any modeling combining the lake budget and glacier mass balance (e.g. Martin et al., 2018), the design of your study does not permit to separate the respective impacts of precipitation and temperature on the glacier-lake changes that you observe.

While we cannot quantify (through modeling) the relative roles of precipitation and temperature, we can state that the climatic cause of the abrupt reductions in lake level and retreat of glaciers in the Junín catchment during MIS 2-3 were not driven primarily by temperature, and the correlation with the Pacupahuain speleothem record confirms reduced monsoon strength during these intervals. In contrast, the complete and final deglaciation of the Junín catchment was driven primarily by warming rather than by reduced precipitation and that is why the ice retreated but the sediment record does not contain peat layers at this time.

Line 216: What is a “small sample mass” and a “low CO₂ gas yield”? Provide numbers.

We added this information on sample mass and CO₂ gas yields (lines 248-249).

Figure 1: It would be useful to add here a regional map of South America with the main atmospheric features.

This is a good suggestion and we inset a regional map of South America in figure 1 showing the locations of the Andean records discussed in the text and the outline of the Amazon drainage basin for reference.

Reviewers' Comments:

Reviewer #1:

Remarks to the Author:

The authors have addressed many of my concerns, and the revised manuscript is improved. In particular, the paper now includes some reference to the Lake Titicaca record, which was the first lake record from tropical South America to posit changes in precipitation and glacial extent in response to DO variation. The manuscript also now acknowledges studies from the Bolivian Altiplano of millennial variability in moisture driven by the SASM. But the text still minimizes citing the Lake Titicaca record, and I cite two specific instances below where this is the case and where a simple change in language and addition of a reference would more clearly acknowledge that record as a precursor. In addition, the authors continue to treat the Si and Ti records as semi-quantitative measures of moisture (see comments on line 168-175 below), and they are not – they are very qualitative indicators. Thus, I recommend a few additional modifications.

Line 31: Lake Junin is not the first record with sufficient resolution (Lake Titicaca had high accumulation rates and thus sufficiently high temporal resolution during the glacial to show millennial variation)– rather Junin is the first record with sufficient chronologic precision to unambiguously link the response of glaciers and lake levels to serial D-O events. I suggest the text be modified accordingly.

Line 50-52: The Lake Titicaca record linked glacial advance and retreat to DO cycles and should be cited here – the age control indeed was not sufficient to be conclusive, and the Lake Junin data set is much more definitive, but the linkage was proposed earlier (quoting from Fritz et al. 2010): “The coarser units were formed during wetter periods with enhanced fluvial discharge of water and sediment and larger quantities of entrained nearshore sediment. These erosion rates were likely further enhanced by precipitation-sensitive glacial advance in the eastern Cordillera (Seltzer, 1992; Kull and Grosjean, 2000).”

Line 154-155: I appreciate the addition of a reference to Lake Titicaca and its record of DO variability.

Line 168-175: The authors have not adequately addressed my concerns here. I agree, as stated in their rebuttal, that their record is “diagnostic of the presence or absence of glaciers in the watershed and of substantially lowered lake levels.” The authors now acknowledge that the Junin clastic sediment record is a relative measure of local moisture variation (line 169), but as a qualitative indicator, there is no reason to expect or assume that a change in Ti or Si scales proportionately to the scale of variation in local moisture. This section clearly assumes that it does, and thus that the magnitude of the Si and Ti records are some sort of quantifiable measure of local moisture (“These observations indicate that the local moisture response at Junín can be disproportional to, and possibly even decoupled from, the $\delta^{18}O$ signal.”). I still think this section needs modification in the language to more precisely explain what the Lake Junin record adds relative to the speleothem data.

Sheri Fritz, University of Nebraska - Lincoln

Reviewer #2:

Remarks to the Author:

Re-review of Woods et al

Please note: The line numbers in the .docx source file vs. the PDF of the same file are different. I

believe that the authors' rebuttal gives line numbers that reference the Word doc, so any line numbers I include below are referencing the Word doc.

Generally, the authors have addressed my concerns. In some cases I feel there is more clarification needed and I discuss those places below.

Age control: The authors have sufficiently addressed my concerns regarding age control by inserting more discussion of age uncertainty during the LLGM and toning down some of the claims that could not be so boldly stated due to age uncertainty.

Glaciogenic flux/lake level relationships:

The text in the rebuttal is helpful, but remember that the target audience of Nature Communications is a broad one, not an audience of reviewers with expertise in this field. So while I appreciate the explanation in the rebuttal letter, the additions to the text in lines 130-134 are technical and detailed so they do not clearly address the concern I raised in my earlier review: "If a non-paleolimnologist were to read this paper there would be confusion about why you cannot look at high Ti during the glacial period and infer high lake levels, but then look at low Ti during the Holocene and infer low lake levels (since the lake level today is quite similar to the highest highstands during the glacial period)." I think one simple sentence of addition would suffice but they should spell it out clearly for a non-paleolimnologist. The auths could even modify the text they wrote in the rebuttal letter, since it is well phrased. See for example their response to my minor comment about TOC: "TOC during the Holocene was governed more by the deposition of organic rich lake sediments likely reflecting in-lake (algal) productivity, and is indicative of a different sedimentary environment than the glacial-age peat layers." That "indicative of a different sedimentary environment than the glacial-age...." part is clear, informative, and sums it up nicely for a broad Nat Comms audience.

Regarding this comment:

- "Glaciogenic sediment input to Lake Junín was especially high from 28.5-22.5 ka, which corresponds to the age of moraines deposited during the maximum extent of ice in the last 50 ka in the adjacent eastern cordillera" (lines 106-109). The timing of the high glaciogenic input is questionable here. All that is clear is that it was high at some point before 22.5 ka and after 28.5 ka. But it is quite possible that sedimentation rates proceeded at their pre-28.5 ka values until 23 ka or 24 ka or any other time within that interval. Assuming that the sed rate changed essentially right after the last 14C date prior to the interval seems unfair.

While we cannot know precisely when the sedimentation rates changed across this interval, we do see that Ti and Si increased abruptly at the beginning of this interval, consistent with accelerated clastic input. Furthermore, the age modelling software that we used in this study (Bacon), does not simply linearly interpolate between dated intervals as other age modeling software packages do. A defining feature of Bacon is that it uses a gamma autoregressive process to model accumulation rates based on prior information to establish a coherent evolution of deposition along the core, whereby the software integrates a "memory" of other dated intervals to allow for a more realistic shape or smoothness of the accumulation rates (Blaauw and Christen, 2010).

>> No amount of Bayesian statistics can make up data where there are none. Particularly during a time interval with likely no analog in the rest of the sediment core. That said, I do feel that the text additions that were made have sufficiently addressed my original concern.

Regarding this comment:

- Line 132-133: Throughout the manuscript and in this sentence the authors refer to "P-E" but here

they are demonstrating that the "E" component is not significant. Why not just state that Lake Junín is especially sensitive to "precipitation amount" ? There are lots of other factors that control E other than temperature (e.g. wind strength) but those do not appear to be within the scope of the manuscript, so it would be simpler to just talk about P.

While Junín appears more sensitive to precipitation amount, we cannot rule out changes in E so it is probably still more accurate to state this as P-E.

>> This does the manuscript a disservice. There is a substantial difference between P and P-E and it's a very important distinction for climate modelers and non-paleolim proxy data generators who will read this paper in search of its climatic ramifications. Most lake records are sensitive to P-E and so when you find one that you can show is more biased towards P than E then that is useful and should be stated clearly. At least state the environmental controls as "P-E, but mainly P." Of course you can never rule out all the factors that contribute to changes in E but that doesn't mean you should not stand by your own interpretation of your data.

Regarding this comment:

- Lines 85-87: Provide a citation for the "less than 7% falling during the winter (JJA)" number
This number is based on unpublished rainfall data (1916-1998 CE) for La Oroya, Peru, from the Servicio Nacional de Meteorología e Hidrología del Perú, Ministeria del Ambiente. It appears from the Nature Communications submission guidelines that data sets can only be cited in the references if they have a DOI, and footnotes are not used, so we defer to the Editor on whether we can cite this information.

>>This is a very important data source, and I encourage the Editor to allow citation. I understand the requirement of a DOI in theory, but in practice, requiring a DOI rules out citing most meteorological datasets from developing countries, which is completely unethical.

Reviewer #3:

Remarks to the Author:

Authors well improved the initial manuscript during the revision process. They now better quote the state of the art, and the majority of my concerns were rather satisfactory addressed. The discussion is quite well-balanced and represent in itself a useful review of the state of the art in the field. I now think that the paper should be close to be accepted for publication. However, I still have one suggestion of modification. This last minor concern I have is about the description of the way 10Be ages were recomputed. To avoid misleading future readers, I strongly recommend that you add few words about the Heyman et al., 2016 did recompute the previous ages of Smith et al., 2005.

The Heyman2016 recalculation uses the LSD (Lifton et al., 2014) scaling and a worldwide averaged production rate, without specifying which atmospheric model was used. These 10Be samples are located in the Tropical Andes, where the production rates sensitivity to the magnetic field is high, and where the atmospheric pressure is different from the standard atmosphere. Without knowing further details, there is thus a risk that the approach of Heyman2016 could be biased (or considered as by readers). To check this, I thus recomputed all the Smith ages using the CREp online calculator (<https://crep.otelo.univ-lorraine.fr/#/>, Martin et al., QG, 2017) using the local production rates of (Martin et al., QG, 2015) and (Kelly et al., QG, 2015). Fortunately, the choice of a local scaling changed the ages recomputed by Heyman2016 by 2% only, meaning that Heyman2016 probably used a distributed atmospheric grid, such as ERA40. In order to make this point clear, and save time to future readers that would be doubtful and tempted to recompute the Smith2005 ages with local

production rates, I recommend that you add a sentence (line 118 or in the Method section) stating something like: "The scaling procedure applied by (Heyman et al., 2016) yields ^{10}Be ages that are, within 2%, compatible with those that would be obtained using local Tropical Andes ^{10}Be production rates (Martin et al., QG, 2015; Kelly et al., QG, 2015)".

REVIEWER COMMENTS

Reviewer #1 (Remarks to the Author):

The authors have addressed many of my concerns, and the revised manuscript is improved. In particular, the paper now includes some reference to the Lake Titicaca record, which was the first lake record from tropical South America to posit changes in precipitation and glacial extent in response to DO variation. The manuscript also now acknowledges studies from the Bolivian Altiplano of millennial variability in moisture driven by the SASM. But the text still minimizes citing the Lake Titicaca record, and I cite two specific instances below where this is the case and where a simple change in language and addition of a reference would more clearly acknowledge that record as a precursor. In addition, the authors continue to treat the Si and Ti records as semi-quantitative measures of moisture (see comments on line 168-175 below), and they are not – they are very qualitative indicators. Thus, I recommend a few additional modifications.

Line 31: Lake Junin is not the first record with sufficient resolution (Lake Titicaca had high accumulation rates and thus sufficiently high temporal resolution during the glacial to show millennial variation)– rather Junin is the first record with sufficient chronologic precision to unambiguously link the response of glaciers and lake levels to serial D-O events. I suggest the text be modified accordingly.

We have replaced the word “resolution” with “chronologic precision” as suggested (line 31).

Line 50-52: The Lake Titicaca record linked glacial advance and retreat to DO cycles and should be cited here – the age control indeed was not sufficient to be conclusive, and the Lake Junin data set is much more definitive, but the linkage was proposed earlier (quoting from Fritz et al. 2010):

“The coarser units were formed during wetter periods with enhanced fluvial discharge of water and sediment and larger quantities of entrained nearshore sediment. These erosion rates were likely further enhanced by precipitation-sensitive glacial advance in the eastern Cordillera (Seltzer, 1992; Kull and Grosjean, 2000).”

The statement quoted above speculates that glacier advance contributed to enhanced erosion rates, but Fritz et al (2010) do not establish a connection between glaciers and the proxies used in the Lake Titicaca record, and the verbiage does not propose a linkage between DO events and glacial advance/retreat. Lake Titicaca offers a record of Andean moisture variations, which we reference and discuss several times in our manuscript (lines 45, 49, 157-158). It is not, however, a record of Andean glacier fluctuations and it would simply be inaccurate to cite it as such.

As we have pointed out in the body of the text and in response to this reviewer’s comments earlier, the glacial connection in the Junín record is well-established and central to our interpretations, unlike that in the Titicaca core. Dozens of cosmogenic exposure ages published by Smith et al (2005) document the connection between moraine building events within

kilometers of the Junín lake shore and clastic flux to Junín. In contrast, the chronology of glaciation in the Bolivian Andes is based primarily on moraines outside the Titicaca drainage basin. It should also be noted that moraines within the Titicaca drainage basin are much further from the lake shore than are those in the Junín basin.

Line 154-155: I appreciate the addition of a reference to Lake Titicaca and its record of DO variability.

Line 168-175: The authors have not adequately addressed my concerns here. I agree, as stated in their rebuttal, that their record is “diagnostic of the presence or absence of glaciers in the watershed and of substantially lowered lake levels.” The authors now acknowledge that the Junin clastic sediment record is a relative measure of local moisture variation (line 169), but as a qualitative indicator, there is no reason to expect or assume that a change in Ti or Si scales proportionately to the scale of variation in local moisture. This section clearly assumes that it does, and thus that the magnitude of the Si and Ti records are some sort of quantifiable measure of local moisture (“These observations indicate that the local moisture response at Junín can be disproportional to, and possibly even decoupled from, the $\delta^{18}O$ signal.”). I still think this section needs modification in the language to more precisely explain what the Lake Junin record adds relative to the speleothem data.

We disagree with the reviewer’s characterization here. We do not treat the Ti and Si records as “some sort of quantifiable measure of local moisture” because we do not apply any calibration or formula to convert the proxy measurement into units of precipitation change. We state that the clastic record provides a record of relative changes in precipitation, which acknowledges that the proxies are qualitative indicators. But as qualitative indicators, they can reveal “relatively dry” versus “very dry” based on their magnitude of change. This is particularly true of the Junín record because it is responding entirely to local moisture changes – there are no upstream effects on the Ti and Si signals like those that affect the isotopic composition of speleothems. Furthermore, as mentioned in our previous rebuttal, the duration of the DO events is legitimately comparable on a quantitative basis, because of the high resolution of data and strong chronologic precision in both the Junín record and the speleothems we compare it to. Thus, we do not feel this section of the manuscript requires further modification.

Sheri Fritz, University of Nebraska - Lincoln

Reviewer #2 (Remarks to the Author):

Re-review of Woods et al

Please note: The line numbers in the .docx source file vs. the PDF of the same file are different. I believe that the authors’ rebuttal gives line numbers that reference the Word doc, so any line numbers I include below are referencing the Word doc.

Generally, the authors have addressed my concerns. In some cases I feel there is more clarification needed and I discuss those places below.

Age control: The authors have sufficiently addressed my concerns regarding age control by inserting more discussion of age uncertainty during the LLGM and toning down some of the claims that could not be so boldly stated due to age uncertainty.

Glaciogenic flux/lake level relationships:

The text in the rebuttal is helpful, but remember that the target audience of Nature Communications is a broad one, not an audience of reviewers with expertise in this field. So while I appreciate the explanation in the rebuttal letter, the additions to the text in lines 130-134 are technical and detailed so they do not clearly address the concern I raised in my earlier review: "If a non-paleolimnologist were to read this paper there would be confusion about why you cannot look at high Ti during the glacial period and infer high lake levels, but then look at low Ti during the Holocene and infer low lake levels (since the lake level today is quite similar to the highest highstands during the glacial period)." I think one simple sentence of addition would suffice but they should spell it out clearly for a non-paleolimnologist. The auths could even modify the text they wrote in the rebuttal letter, since it is well phrased. See for example their response to my minor comment about TOC: "TOC during the Holocene was governed more by the deposition of organic rich lake sediments likely reflecting in-lake (algal) productivity, and is indicative of a different sedimentary environment than the glacial-age peat layers." That "indicative of a different sedimentary environment than the glacial-age...." part is clear, informative, and sums it up nicely for a broad Nat Comms audience.

We appreciate this suggestion to make the text more accessible for the intended audience. We have modified lines 131-133 to better clarify that the cessation of clastic inputs after 20 ka does not imply lower lake level but rather indicates a difference in sedimentary environment.

Regarding this comment:

- "Glaciogenic sediment input to Lake Junín was especially high from 28.5-22.5 ka, which corresponds to the age of moraines deposited during the maximum extent of ice in the last 50 ka in the adjacent eastern cordillera" (lines 106-109). The timing of the high glaciogenic input is questionable here. All that is clear is that it was high at some point before 22.5 ka and after 28.5 ka. But it is quite possible that sedimentation rates proceeded at their pre-28.5 ka values until 23 ka or 24 ka or any other time within that interval. Assuming that the sed rate changed essentially right after the last 14C date prior to the interval seems unfair.

While we cannot know precisely when the sedimentation rates changed across this interval, we do see that Ti and Si increased abruptly at the beginning of this interval, consistent with accelerated clastic input. Furthermore, the age modelling software that we used in this study (Bacon), does not simply linearly interpolate between dated intervals as other age modeling software packages do. A defining feature of Bacon is that it uses a gamma autoregressive process to model accumulation rates based on prior information to establish a coherent evolution of deposition along the core, whereby the software integrates a "memory" of other dated intervals to allow for a more realistic shape or smoothness of the accumulation rates

(Blaauw and Christen, 2010).

>> No amount of Bayesian statistics can make up data where there are none. Particularly during a time interval with likely no analog in the rest of the sediment core. That said, I do feel that the text additions that were made have sufficiently addressed my original concern.

We are glad the text additions have sufficiently addressed the reviewer's concern here.

Regarding this comment:

- Line 132-133: Throughout the manuscript and in this sentence the authors refer to "P-E" but here they are demonstrating that the "E" component is not significant. Why not just state that Lake Junín is especially sensitive to "precipitation amount" ? There are lots of other factors that control E other than temperature (e.g. wind strength) but those do not appear to be within the scope of the manuscript, so it would be simpler to just talk about P.

While Junín appears more sensitive to precipitation amount, we cannot rule out changes in E so it is probably still more accurate to state this as P-E.

>> This does the manuscript a disservice. There is a substantial difference between P and P-E and it's a very important distinction for climate modelers and non-paleolim proxy data generators who will read this paper in search of its climatic ramifications. Most lake records are sensitive to P-E and so when you find one that you can show is more biased towards P than E then that is useful and should be stated clearly. At least state the environmental controls as "P-E, but mainly P." Of course you can never rule out all the factors that contribute to changes in E but that doesn't mean you should not stand by your own interpretation of your data.

Without some sort of hydrological modeling component we can't firmly say whether the lake itself is more biased towards P or E, but oxygen isotope data from Lake Junín compared to nearby open basin lakes during the Holocene confirms that the lake water is sensitive to E (e.g. Bird et al 2011 EPSL), so it would be misleading to state otherwise.

It is probably accurate to say that the proxies for clastic inputs are more biased towards responding to changes in P than to changes in E. The peat layers as a proxy for lake level are likely somewhat more sensitive to E than the clastic proxies are, but it's hard to say whether the peat layers themselves are relatively more sensitive to P versus E as it is clearly a combination of the two. That said, we have removed two references to P-E where the term precipitation would be more appropriate (line 152 and line 201).

Regarding this comment:

- Lines 85-87: Provide a citation for the "less than 7% falling during the winter (JJA)" number. This number is based on unpublished rainfall data (1916-1998 CE) for La Oroya, Peru, from the Servicio Nacional de Meteorología e Hidrología del Perú, Ministeria del Ambiente. It appears from the Nature Communications submission guidelines that data sets can only be cited in the

references if they have a DOI, and footnotes are not used, so we defer to the Editor on whether we can cite this information.

>>This is a very important data source, and I encourage the Editor to allow citation. I understand the requirement of a DOI in theory, but in practice, requiring a DOI rules out citing most meteorological datasets from developing countries, which is completely unethical.

We agree with the reviewer and also encourage the Editor to allow citation of this data source.

Reviewer #3 (Remarks to the Author):

Authors well improved the initial manuscript during the revision process. They now better quote the state of the art, and the majority of my concerns were rather satisfactorily addressed. The discussion is quite well-balanced and represents in itself a useful review of the state of the art in the field. I now think that the paper should be close to being accepted for publication. However, I still have one suggestion of modification. This last minor concern I have is about the description of the way ^{10}Be ages were recomputed. To avoid misleading future readers, I strongly recommend that you add a few words about the Heyman et al., 2016 did recompute the previous ages of Smith et al., 2005.

The Heyman2016 recalculation uses the LSD (Lifton et al., 2014) scaling and a worldwide averaged production rate, without specifying which atmospheric model was used. These ^{10}Be samples are located in the Tropical Andes, where the production rates sensitivity to the magnetic field is high, and where the atmospheric pressure is different from the standard atmosphere. Without knowing further details, there is thus a risk that the approach of Heyman2016 could be biased (or considered as such by readers). To check this, I thus recomputed all the Smith ages using the CREP online calculator (<https://crep.otelo.univ-lorraine.fr/#/>), Martin et al., QG, 2017) using the local production rates of (Martin et al., QG, 2015) and (Kelly et al., QG, 2015). Fortunately, the choice of a local scaling changed the ages recomputed by Heyman2016 by 2% only, meaning that Heyman2016 probably used a distributed atmospheric grid, such as ERA40. In order to make this point clear, and save time to future readers that would be doubtful and tempted to recompute the Smith2005 ages with local production rates, I recommend that you add a sentence (line 118 or in the Method section) stating something like: "The scaling procedure applied by (Heyman et al., 2016) yields ^{10}Be ages that are, within 2%, compatible with those that would be obtained using local Tropical Andes ^{10}Be production rates (Martin et al., QG, 2015; Kelly et al., QG, 2015)".

We thank the reviewer for this helpful analysis of the ^{10}Be ages and we have added the suggested text to the manuscript (lines 117-119).

Reviewers' Comments:

Reviewer #1:

Remarks to the Author:

I had 2 criticisms of the last version of this manuscript and will accept Abbott's rebuttal. I still don't agree with him about the issue of scaling of the isotopic and sedimentological signals, but the community can evaluate the arguments for themselves. It's a high quality record that deserves to be published and discussed.

Reviewer #2:

Remarks to the Author:

Originally, this manuscript shared some unique data and made some exciting claims but some of these claims were overstated or the interpretation was not clear. Now, after the revisions, the manuscript now showcases this unique dataset with a much more justifiable and careful interpretation that will make it much more useful and impactful. While I do not agree with every interpretation in this paper, the authors have addressed my comments sufficiently and I also believe they have now taken the other reviewers' concerns into fair consideration. They argue their claims well and I think the field will really benefit from this contribution. I look forward to seeing this published.

Reviewer #3:

Remarks to the Author:

Authors have considered my last suggestion and I think that the article is now close to be publishable.

I noticed that authors disagree with the Reviewer1's suggestion to quote the previous work of (Fritz et al., QSR, 2010) about the link between the Lake Titicaca sedimentary record and the impact of local glaciers fluctuations on this record during the D/O events. If editor wish to know my external and neutral opinion about this disagreement, here it is: I agree with the authors statement (in their rebuttal letter) that the glacier chronology in the Titicaca watershed is less firm than the one of the glaciers of the Junin area. However, I also think that Fritz et al, indeed proposed 10 years ago (QSR, 2010) a reasonable explanation to link glacier fluctuations and the Titicaca record and that they can be acknowledged for this. In their 2010 paper, their statement is more than speculative "verbiage", it is a working and parsimonious hypothesis that deserved more precise data to be tested. To find a reasonable compromise, I suggest that authors could, for example, revise lines 50-52 this way: "Less is known about the effects of the shorter duration DO cycles on precipitation anomalies and on the mass balance of tropical Andean glaciers. Although some studies proposed a causal link between local glaciers fluctuations and the sedimentary records of Lake Titicaca (Fritz et al., QSR, 2010), well-dated continuous records are still necessary to test/confirm this hypothesis."

REVIEWERS' COMMENTS:

Reviewer #1 (Remarks to the Author):

I had 2 criticisms of the last version of this manuscript and will accept Abbott's rebuttal. I still don't agree with him about the issue of scaling of the isotopic and sedimentological signals, but the community can evaluate the arguments for themselves. It's a high quality record that deserves to be published and discussed.

Reviewer #2 (Remarks to the Author):

Originally, this manuscript shared some unique data and made some exciting claims but some of these claims were overstated or the interpretation was not clear. Now, after the revisions, the manuscript now showcases this unique dataset with a much more justifiable and careful interpretation that will make it much more useful and impactful. While I do not agree with every interpretation in this paper, the authors have addressed my comments sufficiently and I also believe they have now taken the other reviewers' concerns into fair consideration. They argue their claims well and I think the field will really benefit from this contribution. I look forward to seeing this published.

Reviewer #3 (Remarks to the Author):

Authors have considered my last suggestion and I think that the article is now close to be publishable.

I noticed that authors disagree with the Reviewer1's suggestion to quote the previous work of (Fritz et al., QSR, 2010) about the link between the Lake Titicaca sedimentary record and the impact of local glaciers fluctuations on this record during the D/O events. If editor wish to know my external and neutral opinion about this disagreement, here it is: I agree with the authors statement (in their rebuttal letter) that the glacier chronology in the Titicaca watershed is less firm than the one of the glaciers of the Junin area. However, I also think that Fritz et al, indeed proposed 10 years ago (QSR, 2010) a reasonable explanation to link glacier fluctuations and the Titicaca record and that they can be acknowledged for this. In their 2010 paper, their statement is more than speculative "verbiage", it is a working and parsimonious hypothesis that deserved more precise data to be tested. To find a reasonable compromise, I suggest that authors could, for example, revise lines

50-52 this way:

"Less is known about the effects of the shorter duration DO cycles on precipitation anomalies and on the mass balance of tropical Andean glaciers. Although some studies proposed a causal link between local glaciers fluctuations and the sedimentary records of Lake Titicaca (Fritz et al., QSR, 2010), well-dated continuous records are still necessary to test/confirm this hypothesis."

- This is a good suggestion and we have included this statement in the final manuscript.